# Overcoming Data and Model heterogeneities in Decentralized Federated Learning via Synthetic Anchors

## Abstract

Conventional Federated Learning (FL) involves collaborative training of a global model by multiple client local models. In this emerging paradigm, the central server assumes a critical role in aggregating local models and maintaining the global model. However, it encounters various challenges, including scalability, management, and inefficiencies arising from idle client devices. Recently, studies on serverless decentralized FL have shown advantages in overcoming these challenges, enabling clients to own different local models and separately optimize local data. Despite the promising advancements in decentralized FL, it is crucial to thoroughly investigate the implications of data and model heterogeneity, which pose unique challenges that must be overcome. Therefore, the research question to be answered in this study is: *How can every client's local model learn generalizable representation?* To address this question, we propose a novel **De**centralized FL technique by introducing **S**ynthetic **A**nchors, dubbed as DeSA. Inspired by the theory of domain adaptation and Knowledge distillation (KD), we leverage the synthetic anchors to design two effective regularization terms for local training: *1) anchor loss* that matches the distribution of the client's latent embedding with an anchor and *2) KD loss* that enables clients learning from others. In contrast to previous KD-based heterogeneous FL methods, we don't presume access to real public or a global data generator. DeSA enables each client's model to become robust to distribution shift across different client-domains. Through extensive experiments on diverse client data distributions, we showcase the effectiveness of DeSA in enhancing both inter and intra-domain accuracy of each client.

## 1 Introduction

Federated learning (FL) has emerged as an important paradigm to perform machine learning from multi-source data in a distributed manner. Conventional FL techniques leverage a large number of clients to process a global model learning, which is coodinated by a central server. Unfortunately, the conventional FL techniques face poor performance due to the presence of heterogeneities. **Data-heterogeneity** involves relaxing the assumption that the data across all the client are independent and identically distributed (i.i.d.). To solve the problem, a plethora of methods have been proposed. However, most of the works handling the data heterogeneity assumes that the model architectures are invariant across clients (Li et al., 2020b;a; 2021b; Karimireddy et al., 2020; Tang et al., 2022). However, many practical FL applications (*e.g.*, Internet-of-Things and mobile device system) face **Model-heterogeneity**, where clients have devices with different computation capabilities and memory constraints. Thereby, it becomes necessary to allow each client to have different model architecture. Since conventional FL methods that require model parameter sharing cannot be applied in this setting, to address the model heterogeneity issue, strategies have been proposed to leverage knowledge transferring, *e.g.*, server collects labeled data with the similar distribution as the client data or clients transmit models (Lin et al., 2020; Zhu et al., 2021). Additionally, these operations usually require a server to coordinate the knowledge distillation.

Besides heterogeneity, there arises concerns on increased vulnerability of system failures and trustworthiness concerns for the central server design in the conventional FL. An emerging paradigm, called **decentralized FL**, is featured by its severless setting to address the issues. Recent work has

shown decentralized FL framework can provide more flexibility and solubility (Beltrán et al., 2023; Yuan et al., 2023b). However, without the use of central server for model aggregation of knowledge transferring, the aforementioned heterogeneous FL methods could not be directly applied to decentralized FL. Furthermore, most of the works in decentralized FL focus on model personalization, deflecting the generalization capability of each client models (Huang et al., 2022). It is crucial for decentralized FL to be generalizable since local training data may not align with local testing data in practice.

We can see that both heterogeneous FL and decentralized FL leave the gray space of the following practical research question: *How can **every** client model perform well on other client domains(generalization), in a completely decentralized heterogeneous FL setup?* Such a problem is referred as *decentralized federated mutual learning*, which is further detailed in Section 2.2.

To the best of our knowledge, we are the first to address both data and model heterogeneity issues under serverless decentralized FL setting (see the comparison with related work in Table 1). In particular, we achieve this by performing local heterogeneity harmonized training and knowledge distillation. To this end, we use a lightweight synthetic data generation process via distribution matching (Zhao & Bilen, 2023). The synthetic data are exchangeable across clients to augment local datasets and serve as **anchor points** to improves FL for two purposes: 1) reducing the domain-gap between the distributions of the learnt features; and 2) enabling local knowledge distillation on predicted logits under model heterogeneity setting. In summary, we tackle a realistic and challeng-

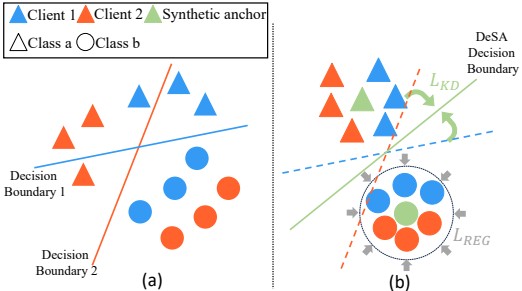

Figure 1: The decision boundary before (a) and after (b) applying our proposed $\mathcal{L}_{\text{REG}}$ (Eq. 4) and $\mathcal{L}_{\text{KD}}$ (Eq. 6) using our synthetic anchor data. $\mathcal{L}_{\text{REG}}$ aims to group the raw feature towards synthetic anchor feature, and $\mathcal{L}_{\text{KD}}$ twists the local decision boundary towards the generalized decision boundary.

ing setting in decentralized FL, where both data and model heterogeneities exist, and our contributions are listed as follows:

- To circumvent the heterogeneity on data and model, we propose an innovative algorithm named Decentralized Federated Learning with Synthetic Anchors (DESA). In DESA, clients communicate directly, eliminating the need for a central server. Unlike other FL frameworks using real data, our approach generates a small synthetic anchor data to enhance client-model generalization.
- To optimize models, our novel FL loss function combines local cross-entropy, synthetic anchor, and cross-client knowledge distillation losses. Our theoretical analysis confirms that a strategic design of synthetic anchor data and correct leveraging of client knowledge boosts local model generalization in diverse data scenarios.
- We conduct extensive experiments prove DESA's effectiveness, surpassing existing decentralized FL algorithms. It excels in inter- and intra-client performance across diverse tasks, even handling data shifts and model differences.

## 2 PRELIMINARIES

### 2.1 CONVENTIONAL FEDERATED LEARNING

Conventional FL aims to learn a **single generalized global model** that performs optimally on all the clients' data domains. Mathematically, the learning problem can be formulated as

$$M^* = \arg \min_{M \in \mathcal{M}} \sum_{i=1}^{N} \mathbb{E}_{\mathbf{x}, y \sim P_i} [\mathcal{L}(M(\mathbf{x}), y)] \tag{1}$$

where $M^*$ is the optimized global model from the shared model space $\mathcal{M}$, $P_i$ is the local data distribution at the $i$th client, $\mathcal{L}$ is the loss function and $(\mathbf{x}, y)$ is the feature-label pair. Inspired by the pioneering work of FedAvg (McMahan et al., 2017), a plethora of methods have tried to fill in the performance gap of FedAvg on data-heterogeneous scenario, which can be categorized in two main orthogonal directions: *Direction 1* aims to minimize the difference between the local and global

Table 1: Comparison of the settings with other related heterogeneous FL and decentralized FL methods.

| Methods | Data Heterogeneity | Model Heterogeneity | Serverless | No Public data | Mutual Learning |
|---|---|---|---|---|---|
| VHL (Tang et al., 2022) | ✓ | ✗ | ✗ | ✓ | ✓[a] |
| FedGen (Zhu et al., 2021) | ✓ | ✗ | ✗ | ✓ | ✓ |
| FedHe (Chan & Ngai, 2021) | ✗ | ✓ | ✓ | ✓ | ✓ |
| FedDF (Lin et al., 2020) | ✓ | ✓ | ✗ | ✗ | ✓ |
| FCCL (Huang et al., 2022) | ✓ | ✓ | ✗ | ✗ | ✓ |
| FedFTG (Zhang et al., 2022b) | ✓ | ✓ | ✗ | ✓ | ✓ |
| DENSE (Zhang et al., 2022a) | ✓ | ✓ | ✗ | ✓ | ✓ |
| DESA (ours) | ✓ | ✓ | ✓ | ✓ | ✓ |

[a] VHL has a single global model, trained using mutual information from all clients. Therefore we reference it under Mutual Learning.

model parameters to improve convergence (Li et al., 2020a; Karimireddy et al., 2020; Wang et al., 2020). *Direction 2* enforces consistency in local embedded features using anchors and regularization loss (Tang et al., 2022; Zhou et al., 2022; Ye et al., 2022). This work follows the second research direction and aim to leverage anchor points to handle data heterogeneity. We also tackle the more challenging problem of domain shift, unlike other methods that only assume a label-shift amongst the client-data distributions.

## 2.2 DECENTRALIZED FL AND MUTUAL LEARNING

Standard decentralized FL aims to solve the same generalization objective as conventional FL (*i.e.*, Eq. 1), only, without a central server to do so (Gao et al., 2022). Here, we focus on learning from each other under heterogeneous models and data distributions. This brings in another line of work, known as *mutual learning*. Although mutual learning with heterogeneous data and models has been studied recently, most of them assume the existence of public real data (Lin et al., 2020; Huang et al., 2022; Gong et al., 2022) or a central server to coordinate the generation of synthetic data from the local client data (Zhang et al., 2022a; Zhu et al., 2021; Zhang et al., 2022b). Another related work, FedHe (Chan & Ngai, 2021) proposes to share averaged information from local data. However, none of the above methods simultaneously address the non-iid features and heterogeneous models issue under serverless and data-free setting. In this work, we explore mutual learning to optimize both local (intra-client) and global (inter-client) dataset accuracy (see the detailed setup in Sec. 3.1). We list the comparison with other methods in Table 1 and more detailed related works in Appendix G.

## 3 METHOD

### 3.1 NOTATION AND PROBLEM SETUP

Suppose there are $N$ clients with $i$th client denoted as $C_i$. Let's represent the **private datasets** on $C_i$ as $D_i = \{\mathbf{x}, y\}$, where $\mathbf{x}$ is the feature and $y \in \{1, \cdots, K\}$ is the label from $K$ classes. Let $\mathcal{L}$ represent a real valued loss function for classification(Cross-entropy loss). Denote the communication neighboring nodes of the client $C_i$ in the system as $\mathcal{N}(C_i)$. We denote the local models as $\{M_i = \rho_i \circ \psi_i\}_{i=1}^{i=N}$, where $\psi_i$ represents the feature encoder and $\rho_i$ represents the classification head for the $i$th client's model $M_i$. DESA returns trained client models $\{M_i\}_{i=1}^{i=N}$.

Our work aims to connect two key areas in heterogeneous FL and decentralized FL and thus it forms the problem of *decentralized federated mutual learning*. Mutual learning is essential in decentralized FL, where, we train **multiple client models** in a decentralized way such that they can **generalize** well across all clients' data domains. Mathematically, our objective is formulated as,

$$\text{For every client } i, \ M_i^* = \underset{M_i \in \mathcal{M}_i}{\arg\min} \underbrace{\mathbb{E}_{\mathbf{x},y\sim P_i}[\mathcal{L}(M_i(\mathbf{x}), y)]}_{\text{Intra-client}} + \underbrace{\sum_{j\in\mathcal{N}(C_i)} \mathbb{E}_{\mathbf{x},y\sim P_j}[\mathcal{L}(M_i(\mathbf{x}), y)]}_{\text{Inter-client}}, \quad (2)$$

where $M_i^*$ is the best possible model for client $i$ with respect to the model space $\mathcal{M}_i$.

**Overview of DESA.** The overall objective of DESA is to improve local models' generalizability in FL training under both model and data heterogeneity in a serverless setting as shown in in Figure 2 a). The pipeline of DESA is depicted in Figure 2 b). Our algorithm contains three important aspects: 1) First, we generate synthetic anchor data that is shared amongst the client's neighbors; 2) Second, we train each client model locally on its own private dataset along with a synthetic anchor based

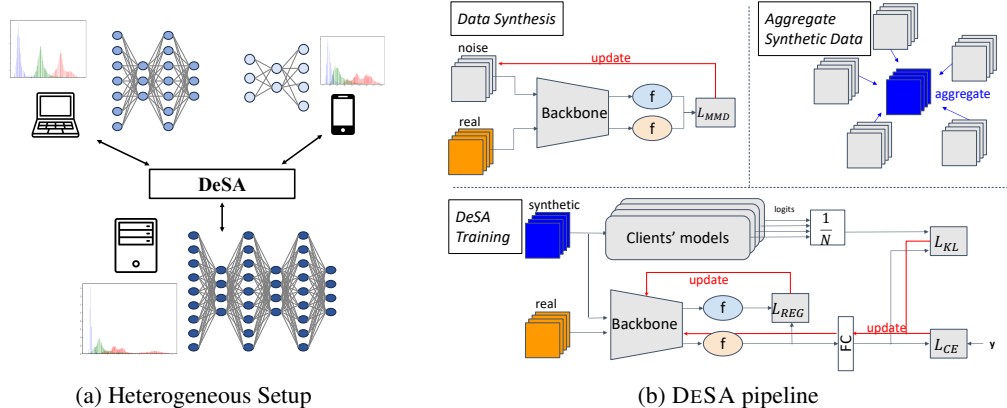

(a) Heterogeneous Setup                    (b) DeSA pipeline

Figure 2: Heterogeneous setup and DeSA pipeline. (a) We assume a realistic FL scenario, where clients have different data distributions and computational powers, which results in different model architectures. (b) DeSA pipeline consists of three phases, local data synthesis (top left), global synthetic data aggregation (top right)(Section 3.2), and decentralized training (bottom) using anchor regularization(Section 3.3) and knowledge distillation (Section 3.4).

regularizer and 3) Third, we allow the models to learn from each other in-order to boost both inter and intra domain performance via knowledge distillation based on the synthetic anchors. Note that during DeSA training, clients only need to share logits *w.r.t.* global synthetic data, resulting in a lightweight communication overhead (as discussed in Appendix D) . The effectiveness of steps 2 and 3 can be observed in Figure 1. The next three subsections delve deeper into these three designs. The full algorithm is depicted in Algorithm 1.

### 3.2 SYNTHETIC ANCHOR DATASETS GENERATION

The recent success of dataset distillation-based data synthesis technique that generates data with similar representation power as the original raw data(Zhao et al., 2020; Zhao & Bilen, 2023). Thus, we propose to leverage this method to efficiently and effectively generate a synthetic anchor dataset without requiring any additional model pretraining. Specifically, we utilize distribution matching (Zhao & Bilen, 2023) to distill local synthetic anchor data using the empirical maximum mean discrepancy loss (MMD) (Gretton et al., 2012) as follows,

$$D_i^{Syn} = \arg\min_D ||\frac{1}{|D_i|} \sum_{(\mathbf{x},y)\in D_i} \psi^{\text{rand}}(\mathbf{x}|y) - \frac{1}{|D|} \sum_{(\mathbf{x},y)\in D} \psi^{\text{rand}}(\mathbf{x}|y)||^2, \tag{3}$$

where $\psi^{\text{rand}}$ is a randomly sampled feature extractor for each iteration, $D_i$ is the raw data for client $i$, and $D_i^{Syn}$ is its target synthetic data.

Similar to other FL work sharing distilled synthetic data for efficiency (Song et al., 2023), we request each client to generate a local synthetic anchor dataset and share it among peers. We denote the synthetic anchor data as $D^{Syn} = \cup_i D_i^{Syn}$ [1]. Following Eq. 3, the synthetic anchor datasets are generated in a **class-balanced** manner, which enables the label prior to being unbiased towards a set of classes. Different from (Song et al., 2023), we further propose novel loss terms and training strategies to help mitigate the distribution discrepancy between the clients, which are detailed in the following sections (see Sec. 3.3 and Sec. 3.3), enabling improved model performance (see the ablation study in Sec. 5.4).

Sharing image-level information among clients may raise privacy concerns. However, we claim that decentralized FL with both data and model heterogeneities is an extremely challenging setting, where existing solutions either require sharing real public data (Lin et al., 2020; Huang et al., 2022) or synthetic data generated from real data with GAN-based generator (Zhang et al., 2022a;b).

---

[1] By default, we perform simple interpolation (averaging) among clients as it is shown that using this mixup strategy can improve model fairness (Chuang & Mroueh, 2021)

Instead, we propose to use distribution matching to distill data, a simple and less data-greedy strategy, for data synthesis. Research has shown that using distilled data can defend against privacy attacks (Dong et al., 2022) such as membership inference attacks (MIA) (Shokri et al., 2017) and gradient inversion attacks (Huang et al., 2021a). We show the DESA's defense against MIA (Carlini et al., 2022a) in Appendix B. In addition, recent papers have successfully applied differential privacy (DP) (Abadi et al., 2016) mechanism into data distillation (Xiong et al., 2023; Wang et al., 2023) to ensure privacy. We also discuss how we add DP into data distillation following (Xiong et al., 2023) and show that DESA is still effective using the DP synthetic anchor data in Appendix C. The potential privacy risk of FL with DESA is beyond the main scope of our study, as this commonly exists in the related work mentioned above and we fairly align with their settings in our comparisons.

## 3.3 ANCHOR LOSS FOR DATA HETEROGENEITY

The synthetic anchor regularization term enforces the model to learn a **client-domain invariant** representation of the data. (Tang et al., 2022) and several other domain adaptation works show that, adding a distribution discrepancy based loss in the **latent space** enables learning of a domain-invariant encoder $\psi$. However, most of the domain adaptation works require explicit access to the data from other domains, or instead use random noise as anchor's in the latent space to pull the representations towards it. We propose using the latent space distribution of the synthetic anchor data $D^{Syn}$ as a synthetic anchor to which the client-model specific encoders $\psi_i$ can project their **local private data** onto. The loss function can be therefore defined as,

$$\mathcal{L}_{REG}(\psi_i) = \sum_{k=1}^{K} \mathbb{E}_{(\mathbf{x},y)\sim D_i, (\mathbf{x}^{Syn},y)\sim D^{Syn}}[d(\psi_i(\mathbf{x}^{Syn})||\psi_i(\mathbf{x}))|y=k], \quad (4)$$

where $K$ is the number of classes , $d$ is distance computed using the supervised contrastive loss,

$$d(\psi_i; D^{Syn}, D_i) = \sum_{j\in B} -\frac{1}{|B_{\setminus j}^{y_j}|} \sum_{\mathbf{x}_p \in B_{\setminus j}^{y_j}} \log \frac{\exp(\psi_i(\mathbf{x}_j) \cdot \psi_i(\mathbf{x}_p)/\tau_{temp})}{\sum_{\mathbf{x}_a \in B_{\setminus j}} \exp(\psi_i(\mathbf{x}_j) \cdot \psi_i(\mathbf{x}_a)/\tau_{temp})} \quad (5)$$

where $B_{\setminus j}$ represents a batch containing both local raw data $D_i$ and global synthetic data $D^{Syn}$ but without data $j$, $B_{\setminus j}^{y_j}$ is a subset of $B_{\setminus j}$ only with samples belonging to class $y_j$, and $\tau_{temp}$ is a scalar temperature parameter. Note that we will detach the synthetic anchor data to ensure we are pulling local features to global features.

## 3.4 KNOWLEDGE DISTILLATION FOR MODEL HETEROGENEITY

This step allows a single client model to learn from all the other models using their predictions on a common synthetic anchor dataset $D^{Syn}$. Under our setting of model heterogeneity among clients, we cannot aggregate the model parameters by simply averaging as in FedAvg (McMahan et al., 2017). Instead, we propose to utilize knowledge distillation for decentralized model aggregation. Specifically, the synthetic anchor dataset is representative of the distributions of the client and its peers together, so minimizing cross entropy on the public dataset $D^{Syn}$ enforces the model to perform well on client-domains that are also not its own. Since the other peer models are already learning using Eq. 4 on their local datasets, we also add a loss that enables the client model to mimic the predictions of the other models on the global dataset $D^{Syn}$.

$$\mathcal{L}_{KD}(M_i) = \mathcal{L}_{KL}(M_i(\mathbf{x}^{Syn}), \bar{Z}_i), \quad \bar{Z}_i = \frac{1}{N-1}\sum_{j\neq i} M_j(\mathbf{x}^{Syn}), \quad (6)$$

where $(\mathbf{x}^{Syn}, y^{Syn}) \sim D^{Syn}$, $\mathcal{L}_{CE}$ is the cross-entropy loss of the synthetic anchor data $\mathbf{x}^{Syn}$ and $\mathcal{L}_{KL}$ is the KL-Divergence between the output logits of $\mathbf{x}_{Syn}$ on $M_i$ and the averaged output logits of $\mathbf{x}^{Syn}$ on $M_j, \forall j \neq i$. As you can see that enforcing the client model $M_i$ to learn on Eq. 6 may reduce the performance of the model on it's own dataset $D_i$. This is because of the phenomenon referred to as **Catastrophic Forgetting**. To prevent this from happening, we take inspiration from **Continual Learning** and we modify the KD loss to also incorporate the standard cross entropy loss on the private dataset $D_i$. This helps the gradients of the losses in both of the steps become coherent. Thus, we formulate the overall loss as

$$\mathcal{L} = \mathcal{L}_{CE}(D_i \cup D^{Syn}; M_i) + \lambda_{REG}\mathcal{L}_{REG}(D_i, D^{Syn}; M_i) + \lambda_{KD}\mathcal{L}_{KD}(D^{Syn}; M_i, \bar{Z}_i), \quad (7)$$

---

**Algorithm 1** Serverless DeSA (Procedures for Client $i$)

---

    **procedure** INIT($C_i$)
        **for all** $j \in \mathcal{N}(C_i)$ **do**          ▷ Communicate with peers to send their synthetic anchor data
            $D^{Syn} = D^{Syn} \cup$ GET-IMG($C_j$)        ▷ Get-img generate synthetic data using Eq. 3.
        **end for**
    **end procedure**
    **procedure** LOCALTRAIN($C_i$, $t$)
        **if** client $C_i$ is sampled **then**                         ▷ Client sampling
            share $Z_i = M_i(D^{Syn})$ to $\mathcal{N}(C_i)$
            get $\bar{Z}_i = 1/|\mathcal{N}(C_i)| \sum_j^{j \in \mathcal{N}(C_i)} Z_j$
            $\mathcal{L}_{CE} =$ CLASSIFICATION($D_i \cup D^{Syn}; M_i$)       ▷ Standard Local Cross entropy
            $\mathcal{L}_{REG} =$ FEATURE-REGULARIZATION($D_i, D^{Syn}; M_i$)     ▷ Anchor Loss with Eq. 4
            $\mathcal{L}_{KD} =$ KD($D^{Syn}; M_i, \bar{Z}$)       ▷ Model knowledge transferring on $D^{Syn}$ with Eq. 6
            $\mathcal{L} = \mathcal{L}_{CE} + \lambda_{REG}\mathcal{L}_{REG} + \lambda_{KD}\mathcal{L}_{KD}$
            $M_i = M_i - \eta\nabla_{M_i}\mathcal{L}$                   ▷ Update local model with Eq. 7
        **end if**
    **end procedure**

---

where $\mathcal{L}_{CE}(D; M) = \frac{1}{|D|}\sum_{\mathbf{x},y \in D} -\sum_k^K [y]_k log([M(\mathbf{x})]_k)$ is the $K$-classes cross entropy loss on data $D$ and model $M$, and $[\cdot]_k$ represents the $k$th element. In Eq. 7, $\lambda_{REG}$ and $\lambda_{KD}$ are the hyperparameters for regularization and KD losses, $\mathcal{L}_{REG}$ and $\mathcal{L}_{KD}$ are as defined in Eq. 4 and Eq. 6, and $\bar{Z}_i$ is the shared logits from other clients.

## 4 THEORETICAL ANALYSIS ON GENERALIZATION

In this section, we focus on providing a theoretical justification for our proposed loss function and models. Consider the $i$th client's local dataset $D_i$ which is sampled from the local data distribution $(\mathbf{x}, y) \sim P_i$. Taking the data heterogeneity into consideration, there exists at least two clients has different local data distributions, i.e. $\exists i \neq j$ s.t. $P_i \neq P_j$. Also, denote the global data distribution as $P^T = \frac{1}{N}\sum_i P_i$ with the labeling function $f^T$: $y = f^T(\mathbf{x}), \forall(\mathbf{x}, y) \sim P^T$. Our ultimate goal is to have each client learn the local model $M_i$, which optimizes the global generalization error $\epsilon_{P^T}(M_i) = \mathbb{P}_{(\mathbf{x},y)\sim P^T}[M_i(\mathbf{x}) \neq y] = \mathbb{P}_{(\mathbf{x})\sim P^T}[M_i(\mathbf{x}) \neq f^T(\mathbf{x})]$. We borrow the notion of $\mathcal{H}\Delta\mathcal{H}$ divergence from Ben-David et al. (2010) that $d_{\mathcal{H}\Delta\mathcal{H}}(D_s, D_T) = \sup_{h,h' \in \mathcal{H}}|\Pr_{x\sim D_s}(h(x) \neq h'(x)) - \Pr_{x\sim D_T}(h(x) \neq h'(x))|$. We also define a constant $\lambda(P_i)$ for each client domain as $\lambda(P_i) = \min_{M \in \mathcal{M}_i} \epsilon_{P_T}(M) + \epsilon_{P_i}(M)$. Furthermore, we assume the global synthetic data $D^{Syn}$ are from the distribution $P^{Syn}$ with the corresponding labeling function $f^{Syn}$. As $D^{Syn}$ is also leveraged for the knowledge distillation, inspired by (Feng et al., 2021), we further denote the extended knowledge distillation (KD) dataset $D_{KD}^{Syn} = \{\mathbf{x}^{Syn}, \frac{1}{N-1}\sum_{i \neq j} M_j(\mathbf{x}^{Syn})\} \sim P_{KD}^{Syn}$, where $M_i(\mathbf{x}^{Syn})$ is the predicted logit on data $\mathbf{x}^{Syn}$ with the $i$th client's model. Obviously, $D_{KD}^{Syn}$ and $D^{Syn}$ shares the same distribution over $\mathbf{x}$, while the labeling function on $D_{KD}^{Syn}$ is $f_{KD}^{Syn}(\mathbf{x}) = \arg\max_c \frac{1}{N-1}\sum_{j \neq i} M_j(\mathbf{x})$, where $c$ is the predicted class. With the defined notation, the following theorem provides the global generalization bound for the $i$th client's model $M_i$.

**Theorem 1.** *Assume the model in $i$-th client $M_i = \rho_i \circ \psi_i$ from the model space $\mathcal{M}_i$. Let the training source data at $i$-th client compose of local data, synthetic data and extended KD data with the component weights $\boldsymbol{\alpha} = [\alpha, \alpha^{Syn}, \alpha_{KD}^{Syn}]^\top$. If $\psi_i \circ P^{Syn} \to \psi_i \circ P^T$ for any $\psi_i$, then the global generalization error can be bounded by*

$$\epsilon_{P^T}(M_i) \leq \epsilon_{P_i^S}(M_i) + \frac{\alpha}{2}d_{\mathcal{H}\Delta\mathcal{H}}(P_i, P^T) + \alpha\lambda(P_i) + \alpha^{Syn}\epsilon_{P^T}(f^{Syn}) + \alpha_{KD}^{Syn}\epsilon_{P^T}(f_{KD}^{Syn}). \quad (8)$$

We provide the interpretation for Theorem 1 in Appendix A.2. Furthermore, the following proposition implies our generalization bound in Theorem 1 is tighter than the original bound under some mild conditions.

**Proposition 2.** *Under the conditions in Theorem 1, if it further holds that*

$$\sup_{M \in \mathcal{M}_i} \min\{|\epsilon_{P^{Syn}}(M) - \epsilon_{P^T}(M)| + \epsilon_{P^T}(f^{Syn}), |\epsilon_{P_{KD}^{Syn}}(M) - \epsilon_{P^T}(M)| + \epsilon_{P^T}(f_{KD}^{Syn})\}$$

$$\leq \inf_{M \in \mathcal{M}_i} |\epsilon_{P_i}(M) - \epsilon_{P^T}(M)| + \frac{1}{2} d_{\mathcal{M}_i \Delta \mathcal{M}_i}(P_i, P^T) + \lambda(P_i), \quad (9)$$

*then we can get a tighter generalization bound on the ith client's model $M_i$ than local learning.*

When the local data heterogeneity is severe, $\inf_{M \in \mathcal{M}_i} |\epsilon_{P_i}(M) - \epsilon_{P^T}(M)|$ and $d_{\mathcal{M}_i \Delta \mathcal{M}_i}(P_i, P^T)$ would be large. As the synthetic data and the extended KD data are approaching the the global data distribution, the left side term in (9) would be small. Thus, the above proposition points out that, to reach better generalization, the model learning should rely more on the synthetic data and the extended KD data, when the local data are highly heterogeneous and the synthetic and the extended KD datasets are similar to the global ones.

## 5 EXPERIMENT

### 5.1 TRAINING SETUP

**Datasets and Models** We extensively evaluate DESA under data heterogeneity in our experiments. Specifically, we consider three classification tasks on three sets of domain-shifted datasets:
1) DIGITS={MNIST (LeCun et al., 1998), SVHN (Netzer et al., 2011), USPS (Hull, 1994), SynthDigits (Ganin & Lempitsky, 2015), MNIST-M (Ganin & Lempitsky, 2015)}, each dataset represents one client.
2) OFFICE={Amazon (Saenko et al., 2010), Caltech (Griffin et al., 2007), DSLR (Saenko et al., 2010), and WebCam (Saenko et al., 2010)}, each dataset represents one client.
3) CIFAR10C consists 57 subsets with domain- and label-shifted datasets sampled with Dirichlet distribution with $\beta = 2$ from Cifar10-C (Hendrycks & Dietterich, 2019).
More information about datasets and image synthesis can be found in Appendix E. In addition to incorporating data heterogeneity, we explicitly showcase that DESA can handle model heterogeneity by setting the models as ConvNet and AlexNet (see Appendix F for model details). In our model heterogeneity experiments (Sec. 5.2), we randomly assign model architectures from {ConvNet, AlexNet} for each client, while in model homogeneous experiments, we use ConvNet for all clients.

**Comparison Methods** We compare DESA with two sets of baseline federated learning methods: one group considers both data and model heterogeneities (Sec. 5.2) and another with homogeneous models(Sec. 5.3). For heterogeneous model experiments, we compare with FedHe (Chan & Ngai, 2021), FedDF (Lin et al., 2020), FCCL (Huang et al., 2022), and VHL (Tang et al., 2022)[2](Deleted FedMD.) . For homogeneous model experiments, we compare with FedAvg (McMahan et al., 2017), FedProx (Li et al., 2020b), MOON (Li et al., 2021b), Scaffold (Karimireddy et al., 2020), FedGen (Zhu et al., 2021), and VHL (Tang et al., 2022).

**FL Training Setup** If not otherwise specified, we use SGD optimizer with a learning rate of $10^{-2}$, and our default setting for local model update epochs is 1, total update rounds is 100, and the batch size for local training is 32. Since we only have a few clients for DIGITS and OFFICE experiments, we will select all the clients for each iteration, while we randomly sample 10% and 20% clients for each round when performing CIFAR10C experiments. By default, $\lambda_{REG}$ and $\lambda_{KD}$ is set to 1.

### 5.2 HETEROGENEOUS MODEL EXPERIMENTS

The experiment results for heterogeneous data and heterogeneous model can be found in Table. 2. The objective of the experiments is to show that DESA can effectively leverage and learn generalized information from other clients even with different model architectures. Thus, we report the intra and inter-client test accuracy by testing $i$-th local model on client $i$'s test set (intra) and every

---

[2]For the purposes of this comparison, we have excluded FedFTG (Zhang et al., 2022b) and DENSE (Zhang et al., 2022a), which address heterogeneities in different learning scenarios. FedFTG focuses on fine-tuning a global model, and DENSE is a one-shot FL, and both of them requires aggregate local information and train a generator on the server side. Note that none of the data-sharing-based baseline methods employ privacy-preserving techniques.

Table 2: Heterogeneous data and heterogeneous model experiments. We compare with baseline FL methods that can handle both data and model heterogeneity on the intra and inter-client test accuracy[a].

| | | | DIGITS | | | | | | OFFICE | | | | | CIFAR10C | |
|---|---|---|---|---|---|---|---|---|---|---|---|---|---|---|---|
| | | | MN(C)[b] | SV(A) | US(C) | Syn(A) | MM(C) | Avg | AM(A) | CA(C) | DS(A) | WE(A) | Avg | 0.1 | 0,2 |
| FedHe | | intra | **98.89** | 83.33 | 98.76 | 93.89 | 94.31 | 93.84 | 53.65 | 52.89 | 25.00 | 88.14 | 54.92 | 48.88 | 55.07 |
| | | inter | 49.67 | 62.50 | 37.67 | 70.77 | 65.89 | 57.30 | 18.12 | 44.72 | 14.10 | 34.82 | 27.94 | 28.35 | 34.82 |
| FedDF | Cifar100 | intra | 92.29 | 20.73 | 86.18 | 15.19 | 51.57 | 53.19 | 10.42 | 12.00 | 15.62 | 15.25 | 13.66 | 60.59 | 62.00 |
| | | inter | 34.85 | 12.39 | 18.28 | 17.21 | 37.28 | 24.00 | 14.29 | 16.67 | 12.56 | 12.68 | 14.05 | 38.17 | 39.38 |
| | FMNIST | intra | 92.59 | 19.56 | 93.33 | 77.86 | 69.83 | 70.63 | 10.42 | 29.33 | 6.25 | 23.73 | 17.43 | 38.52 | 47.24 |
| | | inter | 39.03 | 18.21 | 30.66 | 54.02 | 54.26 | 39.24 | 7.43 | 20.43 | 11.22 | 15.71 | 13.71 | 21.37 | 27.33 |
| FCCL | Cifar100 | intra | 11.25 | 19.10 | 13.66 | 10.08 | 11.25 | 13.07 | 57.81 | 24.44 | 50.50 | 33.90 | 41.54 | **65.14** | **64.89** |
| | | inter | 13.52 | 11.56 | 12.92 | 13.82 | 13.52 | 13.07 | 24.22 | 31.78 | 21.75 | 25.72 | 25.87 | **53.67** | **53.30** |
| | FMNIST | intra | 95.01 | 72.92 | 94.73 | 86.11 | 90.10 | 87.78 | 65.10 | **68.37** | 31.25 | 52.54 | 46.89 | 13.51 | 13.51 |
| | | inter | 34.29 | 58.04 | 29.62 | 57.28 | 60.46 | 47.94 | 20.64 | 29.70 | 15.00 | 22.80 | 22.03 | 12.48 | 12.48 |
| DeSA($D_{\mathrm{VHL}}^{Syn}$)[d] | | intra | 98.69 | 80.09 | 98.66 | 90.68 | 94.40 | 92.50 | 10.42 | 57.33 | 28.13 | **89.83** | 46.43 | 47.99 | 54.42 |
| | | inter | 43.33 | 57.51 | 28.26 | 62.02 | 67.69 | 51.76 | 8.28 | 46.20 | 13.16 | 35.56 | 25.80 | 28.18 | 34.86 |
| DeSA | | intra | 98.78 | **86.42** | **98.87** | **95.21** | **94.54** | **94.77** | **71.88** | 60.44 | **78.13** | **89.83** | **75.07** | 54.06 | 58.14 |
| | | inter | **65.59** | **80.14** | **64.14** | **78.97** | **69.98** | **71.76** | **42.24** | **57.84** | **33.84** | **41.98** | **43.98** | 34.42 | 39.01 |

[a] Intra and inter-client test accuracy are defined as testing $i$-th local model on client $i$'s test set and every client $j$'s ($\forall j \neq i, j \in [N]$) test sets, respectively.
[b] The letter inside the parenthesis is the model architecture used by the client. A and C represent AlexNet and ConvNet, respectively.
[c] The best accuracy is marked in **bold**.
[d] For VHL baseline, we use the synthetic data sampling strategy in VHL only. The purpose is to show DeSA can generate better synthetic anchor data for feature regularization and knowledge distillation.

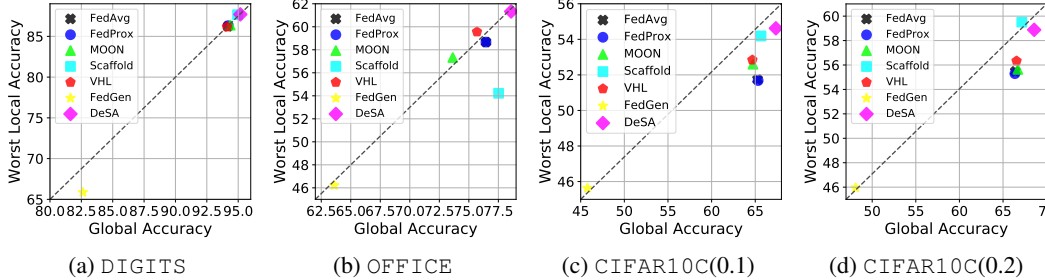

| (a) DIGITS | (b) OFFICE | (c) CIFAR10C(0.1) | (d) CIFAR10C(0.2) |
|---|---|---|---|

Figure 3: Heterogeneous data and homogeneous model experiments. We compare with baseline data heterogeneity FL methods with the same client models. To show that DeSA can effectively leverage information from other clients, we report the averaged global accuracy and the worst test accuracy on local test sets.

client $j$'s ($\forall j \neq i, j \in [N]$) test sets (inter). To fairly compare with FedDM and FCCL, which require accessing to public available data, we test on using Cifar100 (Krizhevsky et al., 2009) and FMNIST (Xiao et al., 2017). DeSA($D_{\mathrm{VHL}}^{Syn}$) uses our training pipeline with synthetic data sampled from an untrained StyleGAN (Karras et al., 2019) as in VHL (Tang et al., 2022).

From the experiments with DIGITS and OFFICE, it is clear that our method DeSA effectively improves both intra- and inter-accuracies. In the CIFAR10C experiments, DeSA improves the intra- and inter-accuracies compared to most of the baseline models, except for FedDF and FCCL with CIFAR100 as the public dataset. We believe this is due to two reasons: 1) CIFAR100 has significantly more data points than our synthetic data, and 2) CIFAR100 and CIFAR10 have a genuine semantic overlap (Krizhevsky et al., 2009), which naturally biases the comparison in favor of methods using additional CIFAR100 information. Nonetheless, it is worth mentioning that the accuracies of FedDF and FCCL drop significantly when switching from Cifar100 to FMNIST. This pattern remains consistent as they perform better with the FMNIST public dataset than using CIFAR100 in the DIGITS experiment. This suggests that the choice of public data for knowledge distillation significantly affects the training outcome of these methods relying on such data. In contrast, our method does not depend on carefully selecting public data.

## 5.3 HOMOGENEOUS MODEL EXPERIMENTS

The objective of the experiments is to show that FedSAB can effectively leverage information from other clients, and further collaboratively train local models under data heterogeneous scenario. Thus, we report the averaged global accuracy and the worst test accuracy on local test sets. One can observe from Figure 3 that DeSA can effectively leverage information from other clients and outperforms other methods in both averaged global accuracies and worst local accuracies. DeSA consistently performs the best regarding the averaged global accuracies. For the worst accuracy, only Scaffold can outperform DeSA in CIFAR10C(0.2) experiment. We believe this is because Scaffold corrects client drift to the server model with the global gradient information on the server side, while DeSA does not require global model or server.

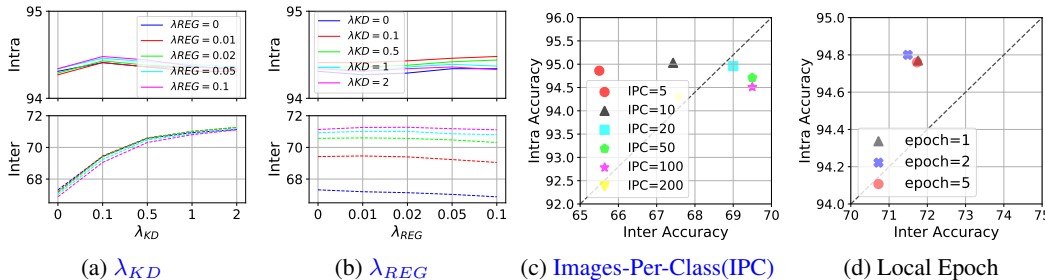

Figure 4: Ablation studies for DESA.

## 5.4 ABLATION STUDIES FOR DESA

The effectiveness of DESA relies on the novel designs of synthetic anchor data and the losses. To evaluate how these designs influences the performance of DESA, we vary the number of synthetic anchor data and the hyperparameters ($\lambda's$) in our loss function. Furthermore, we will discuss how the size of synthetic dataset affects the performance. Lastly, we show that DESA is robust under different local epochs. The ablation studies are run on DIGITS and compare with default hyperparameters under model heterogeneity setting, and we report the averaged intra- and inter-accuracies.

**Evaluation of $\lambda$ Selections.** $\lambda_{KD}$ and $\lambda_{REG}$ play an important role to help the clients to learn generalized information as well as improving the performance on local data. We vary $\lambda_{KD}$ between 0 to 1 along with $\lambda_{REG}$ different selections as shown in Figure 4(a). One can observe that when $\lambda_{KD}$ increases, the inter-accuracy increases. In Figure 4(b), we change $\lambda_{REG}$ between 0 to 0.1 and show that and $\lambda_{REG}$ helps improve the intra-accuracy as well as the inter-accuracy within a certain range (observe the peak inter-accuracy at $\lambda_{REG} = 0.01$). Overall, $\lambda_{KD}$ helps the local model learn information from other clients' models, and $\lambda_{REG}$ improves the local performance by enforcing the local model to learn generalized features.

**Evaluation of Size of Synthetic Dataset.** The size of synthetic data is a critical hyperparameter for DESA as it represents the shared local information. Since DESA synthesizes class-balanced data, we use Images-Per-Class (IPC) to represent the size of the synthetic data. One can observe in Figure 4(c) that blindly increasing the IPC does not guarantee to obtain optimal intra- and inter-accuracy. It will cause the loss function to be dominated by the last 2 terms of Eq. 8, *i.e.,* by synthetic data. However, synthesizing larger number of synthetic data may degrade its quality, and the sampled batch for $\mathcal{L}_{REG}$ may fail to capture the distribution.

**Evaluation of Local Epoch.** Here we present the effect of local epochs on DESA. To ensure fair comparison, we fix the total training iterations for the three experiments, *i.e.,*we set FL communication rounds to 50 when local epochs is 2 to match up with total model updating iterations. Figure 4(d) shows that DESA is robust to various local epoch selections.

## 6 CONCLUSION

A novel and effective method, DESA, is presented that utilizes synthetic data to deal with both data and model heterogeneities in serverless decentralized FL. In particular, DESA introduces a pipeline that involves synthetic data generalization, and we propose a new scheme that incorporates the synthetic data as anchor points in decentralized FL model training. To address heterogeneity issues, we utilize the synthetic anchor data and propose two regularization losses: anchor loss and knowledge distillation loss. We provide theoretical analysis on the generalization bound to justify the effectiveness of DESA using the synthetic anchor data. Empirically, the resulted client models not only achieve compelling local performance but also can generalize well onto other clients' data distributions, boosting inter and intra-domain performance. Through extensive experiments on various classification tasks, we show that DESA robustly improves the efficacy of collaborative learning when compared with state-of-the-art methods, under both model and data heterogeneous settings.

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

**Road Map of Appendix** Our appendix is organized into five sections. The theoretical analysis and proof is in Appendix A. Appendix B shows the results for Membership Inference Attack (MIA) on DESA trained models using DIGITS datasets. Appendix C discusses how we inject DP mechanism in our data synthesis process, and shows that using DP synthetic anchor data for DESA can still yeilds comparable performance. Appendix E introduce the selected datasets and how we synthesize anchor data in detail. Appendix F describes the model architectures (ConvNet and AlexNet) we use in our experiments. Finally, Appendix G provides a detailed literature review about the related works. Our code and model checkpoints are available along with the supplementary materials.

# A  THEORETICAL ANALYSIS AND PROOFS

## A.1  PROOF FOR THEOREM 1

*Proof.* The training data at $i$th client are from as three distributions: 1) the local source data; 2) the global virtual data; 3) the extended KD data. The data from first two groups are used to construct the cross-entropy loss and those from the third one is for the knowledge distillation loss. Without loss of generality, at $i$th client, we set the weight for $P_i$, $P^{Syn}$ and $P_{KD}^{Syn}$ as $\alpha$, $\alpha^{Syn}$ and $\alpha_{KD}^{Syn}$, respectively. For notation simplicity, we assume $\alpha + \alpha^{Syn} + \alpha_{KD}^{Syn} = 1$. Then the training source data at $i$th client is $P_i^S = \alpha P_i + \alpha^{Syn} P^{Syn} + \alpha_{KD}^{Syn} P_{KD}^{Syn}$.

From Theorem 2 in Ben-David et al. (2010), it holds that

$$\epsilon_{P^T}(M_i) \le \epsilon_{P_i}(M_i) + \frac{1}{2} d_{\mathcal{M}_i \Delta \mathcal{M}_i}(P_i, P^T) + \lambda(P_i) \tag{10}$$

where $\frac{1}{2} d_{\mathcal{M}_i \Delta \mathcal{M}_i}(P_i, P^T) = \sup_{M, M' \in \mathcal{M}_i} |\mathbb{P}_{\mathbf{x} \sim P_i}[M(\mathbf{x}) \ne M'(\mathbf{x})] - \mathbb{P}_{\mathbf{x} \sim P^T}[M(\mathbf{x}) \ne M'(\mathbf{x})]|$ and $\lambda(P_i) = \min_{M \in \mathcal{M}_i} \epsilon_{P_i}(M) + \epsilon_{P^T}(M)$ is a constant. Then with (10) and Lemma 1, we have that

$$\begin{aligned}
\epsilon_{P^T}(M_i) \le & \epsilon_{P_i^S}(M_i) + \frac{\alpha}{2} d_{\mathcal{H} \Delta \mathcal{H}}(P_i, P^T) + \alpha_i \lambda(P_i) + \alpha^{Syn} (\sup_{\rho, \rho'} |\epsilon_{\psi \circ P^T}(\rho, \rho') - \epsilon_{\psi \circ P^{Syn}}(\rho, \rho')| \\
& + \epsilon_{P^T}(f^{Syn})) + \alpha_{KD}^{Syn} (\sup_{\rho, \rho'} |\epsilon_{\psi \circ P^T}(\rho, \rho') - \epsilon_{\psi \circ P^{Syn}}(\rho, \rho')| + \epsilon_{P_{KD}^T}(f^{Syn})) \\
\le & \epsilon_{P_i^S}(M_i) + \frac{\alpha}{2} d_{\mathcal{H} \Delta \mathcal{H}}(P_i, P^T) + \alpha \lambda(P_i) + \alpha^{Syn} \epsilon_{P^T}(f^{Syn}) + \alpha_{KD}^{Syn} \epsilon_{P_{KD}^T}(f^{Syn}) \\
& + (\alpha^{Syn} + \alpha_{KD}^{Syn}) \sup_{\rho, \rho'} |\epsilon_{\psi \circ P^T}(\rho, \rho') - \epsilon_{\psi \circ P^{Syn}}(\rho, \rho')| \tag{11}
\end{aligned}$$

With the condition that $\psi \circ P^{Syn} \to \psi \circ P^T$, the above bound can be simplified as

$$\epsilon_{P^T}(M_i) \le \epsilon_{P_i^S}(M_i) + \frac{\alpha}{2} d_{\mathcal{H} \Delta \mathcal{H}}(P_i, P^T) + \alpha \lambda(P_i) + \alpha^{Syn} \epsilon_{P^T}(f^{Syn}) + \alpha_{KD}^{Syn} \epsilon_{P^T}(f_{KD}^{Syn}). \tag{12}$$

$\square$

## A.2  INTERPRETATION FOR THEOREM 1

From Eq. (8), it can be seen that the generalization bound for $M_i$ consists of five terms and relies on a single assumption. The assumption holds true because of the domain invariance imposed by our anchor loss in equation 4.

- The first term $\epsilon_{P_i^S}(h)$ is the error bound with respect to the training source data distribution. With Claim 1 in appendix, minimizing this term is equivalent to optimizing the loss $\alpha \mathbb{E}_{(\mathbf{x}, y) \sim P_i} \mathcal{L}_{CE} + \alpha^{Syn} \mathbb{E}_{(\mathbf{x}, y) \sim P^{Syn}} \mathcal{L}_{CE} + \alpha_{KD}^{Syn} \mathbb{E}_{(\mathbf{x}, y) \sim P^{Syn}} \mathcal{L}_{KD}$
- The second and third terms are inherited from the original generalization bound in Ben-David et al. (2010) with the local training data. For our case, it can be controlled by the component weight $\alpha$. If we rely less on the local data (i.e. smaller $\alpha$), then these terms will be vanishing.

- The fourth term is to measure the discrepancy between real labeling and the synthetic data labeling mechanisms. This discrepancy will be low because of our synthetic data generation process. The distributions of real and synthetic data are matched using MMD in Equation 3. Therefore, the synthetic data labelling $f^{syn}$ will be similar to the real labelling $f_T$, and the error $\epsilon_{P_T}(f^{syn})$ will be minimized.

- The fifth term originates from the knowledge distillation loss in equation 6. Here, we use the consensus knowledge from neighbour models to improve the local model. The labelling function of the extended KD data $f_{KD}^{syn}$, changes as training continues and the neighbour models learn to generalize well. Towards the end of training, predictions from the consensus knowledge should match the predictions of the true labeling function, therefore, $f_{KD}^{syn}$ will be close to $f_T$

**Remark:** In order to get a tight generalization guarantee, we only need one of the fourth or fifth terms to be small. Since, if either any one of them is small, we can adjust the component weights $\alpha$ (practically $\lambda_{REG}$ and $\lambda_{KD}$) to get a better generalization guarantee.

### A.3 PROOF FOR PROPOSITION 2

*Proof.* Without loss of generality, let's start with

$$\sup_{M \in \mathcal{M}_i} |\epsilon_{P^{Syn}}(M) - \epsilon_{P^T}(M)| + \epsilon_{P^T}(f^{Syn}) \leq$$
$$\inf_{M \in \mathcal{M}_i} |\epsilon_{P_i}(M) - \epsilon_{P^T}(M)| + \frac{1}{2}d_{\mathcal{M}_i \Delta \mathcal{M}_i}(P_i, P^T) + \lambda(P_i). \tag{13}$$

Then it holds that for any $M \in \mathcal{M}_i$,

$$\epsilon_{P^{Syn}}(M) - \epsilon_{P^T}(M) + \epsilon_{P^T}(f^{Syn}) \leq \epsilon_{P_i}(M) - \epsilon_{P^T}(M) + \frac{1}{2}d_{\mathcal{M}_i \Delta \mathcal{M}_i}(P_i, P^T) + \lambda(P_i)$$

$$\Rightarrow \quad \epsilon_{P^{Syn}}(M) + \epsilon_{P^T}(f^{Syn}) \leq \epsilon_{P_i}(M) + \frac{1}{2}d_{\mathcal{M}_i \Delta \mathcal{M}_i}(P_i, P^T) + \lambda(P_i) \tag{14}$$

Note that the right side of (14) is the original bound in Theorem 2 in Ben-David et al. (2010). Similarly, we can achieve

$$\epsilon_{P_{KD}^{Syn}}(M) + \epsilon_{P^T}(f_{KD}^{Syn}) \leq \epsilon_{P_i}(M) + \frac{1}{2}d_{\mathcal{M}_i \Delta \mathcal{M}_i}(P_i, P^T) + \lambda(P_i) \tag{15}$$

Combining (14-15) together, we can conclude that our global generalization bound is tighter than the original bound. □

### A.4 SOME USEFUL LEMMAS AND CLAIMS

**Lemma 1.** *Denote the model as $M = \rho \circ \psi \in \mathcal{M}$. The global generalization bound holds as*

$$\epsilon_{P^T}(M) \leq \epsilon_P(M) + \sup_{\rho, \rho'} |\epsilon_{\psi \circ P^T}(\rho, \rho') - \epsilon_{\psi \circ P}(\rho, \rho')| + \epsilon_{P^T}(f), \tag{16}$$

*where $(P, f)$ could be either $(P^{Syn}, f^{Syn})$ or $(P_{KD}^{Syn}, f_{KD}^{Syn})$ pair.*

*Proof.* For any model $M = \rho \circ \psi \in \mathcal{M}$, we have the following bound for the global virtual data distribution:

$$\epsilon_{P^T}(M) - \epsilon_{P^{Syn}}(M) \overset{(a)}{=} \epsilon_{P^T}(M, f^T) - \epsilon_{P^{Syn}}(M, f^{Syn})$$
$$\overset{(b)}{\leq} |\epsilon_{P^T}(M, f^{Syn}) + \epsilon_{P^T}(f^{Syn}, f^T) - \epsilon_{P^{Syn}}(M, f^{Syn})|$$
$$\leq |\epsilon_{P^T}(M, f^{Syn}) - \epsilon_{P^{Syn}}(M, f^{Syn})| + \epsilon_{P^T}(f^{Syn})$$
$$= |\epsilon_{P^T}(\rho \circ \psi, f^{Syn}) - \epsilon_{P^{Syn}}(\rho \circ \psi, f^{Syn})| + \epsilon_{P^T}(f^{Syn})$$
$$= |\epsilon_{\psi \circ P^T}(\rho, f^{Syn} \circ \psi^{-1}) - \epsilon_{\psi \circ P^{Syn}}(\rho, f^{Syn} \circ \psi^{-1})| + \epsilon_{P^T}(f^{Syn})$$
$$\leq \sup_{\rho, \rho'} |\epsilon_{\psi \circ P^T}(\rho, \rho') - \epsilon_{\psi \circ P^{Syn}}(\rho, \rho')| + \epsilon_{P^T}(f^{Syn}) \tag{17}$$

where (a) is by definitions and (b) relies on the triangle inequality for classification error Ben-David et al. (2006); Crammer et al. (2008). Thus, we have that

$$\epsilon_{PT}(M) \leq \epsilon_{P^{Syn}}(M) + \sup_{\rho, \rho'} |\epsilon_{\psi \circ PT}(\rho, \rho') - \epsilon_{\psi \circ P^{Syn}}(\rho, \rho')| + \epsilon_{PT}(f^{Syn}). \qquad (18)$$

Similarly, as the the extended KD dataset shares the same feature distribution with the global virtual dataset, thus the above bound also holds for $f_{KD}^{Syn}$. $\qquad \square$

**Lemma 2** (Appendix A Feng et al. (2021)). *For the extended source domain $(\mathbf{x}^{Syn}, \hat{y}^{Syn}) \sim \hat{P}^{Syn}$, training the related model with the knowledge distillation loss $L_{KD} = D_{KD}(\hat{y}^{Syn} \| h(\mathbf{x}))$ equals to optimizing the task risk $\epsilon_{\hat{P}^{Syn}} = \mathbb{P}_{(\mathbf{x}^{Syn}, \hat{y}^{Syn}) \sim \hat{P}^{Syn}}[h(\mathbf{x}) \neq \arg\max \hat{y}^{Syn}]$.*

**Claim 1.** *With the training source data at $i$th client as $P_i^S$ with the component weight $\boldsymbol{\alpha} = [\alpha, \alpha^{Syn}, \alpha_{KD}^{Syn}]^\top$ on the local data, virtual data and extended KD data, $\epsilon_{P_i^S}(h)$ is minimized by optimizing the loss:*

$$\min_{M \in \mathcal{M}} \alpha \mathbb{E}_{(\mathbf{x},y) \sim P_i} L_{CE}(y, M(\mathbf{x})) + \alpha^{Syn} \mathbb{E}_{(\mathbf{x},y) \sim P^{Syn}} L_{CE}(y, M(\mathbf{x})) + \alpha_{KD}^{Syn} \mathbb{E}_{(\mathbf{x},y) \sim P_{KD}^{Syn}} L_{KL}(y \| M(\mathbf{x})) \tag{19}$$

*Proof.* Note that

$$\min_{M \in \mathcal{M}} \mathbb{E}_{(\mathbf{x},y) \sim P_i^{(S)}} L_{KL}(y \| M(\mathbf{x}))$$

$$\propto \min_{M \in \mathcal{M}} \alpha \mathbb{E}_{(\mathbf{x},y) \sim P_i} L_{KL}(y \| M(\mathbf{x})) + \alpha^{Syn} \mathbb{E}_{(\mathbf{x},y) \sim P^{Syn}} L_{KL}(y \| M(\mathbf{x})) + \alpha_{KD}^{Syn} \mathbb{E}_{(\mathbf{x},y) \sim P_{KD}^{Syn}} L_{KL}(y \| M(\mathbf{x}))$$

$$\overset{(a)}{\propto} \min_{M \in \mathcal{M}} \alpha \mathbb{E}_{(\mathbf{x},y) \sim P_i} L_{CE}(y, M(\mathbf{x})) + \alpha^{Syn} \mathbb{E}_{(\mathbf{x},y) \sim P^{Syn}} L_{CE}(y, M(\mathbf{x})) + \alpha_{KD}^{Syn} \mathbb{E}_{(\mathbf{x},y) \sim P_{KD}^{Syn}} L_{KL}(y \| M(\mathbf{x}))$$

where (a) is because $L_{KL}(y \| h(\mathbf{x})) = L_{CE}(y, h(\mathbf{x})) - H(y)$, where $H(y) = -y \log(y)$ is a constant depending on data distribution. With Lemma 2 and Pinsker's inequality, it is easy to show that $\epsilon_{P_i^S}(h)$ is minimized by optimizing the above loss. $\qquad \square$

## B    MEMBERSHIP INFERENCE ATTACK

We show what under the basic setting of DESA (*i.e.*, not applying Differential Privacy when generating local synthetic data), we can better protect the membership information of local real data than local training or FedAvg (McMahan et al., 2017) on local real data only when facing Membership Inference Attack (MIA) on trained local models. Although we share the logits during communication, it's important to note that these logits are from synthetic anchor data and not real data that needs protection. Therefore, we cannot use MIA methods that rely on logits. Instead, we perform a strong MIA attack recently proposed and evaluate it following the approach in Carlini et al. (2022a).

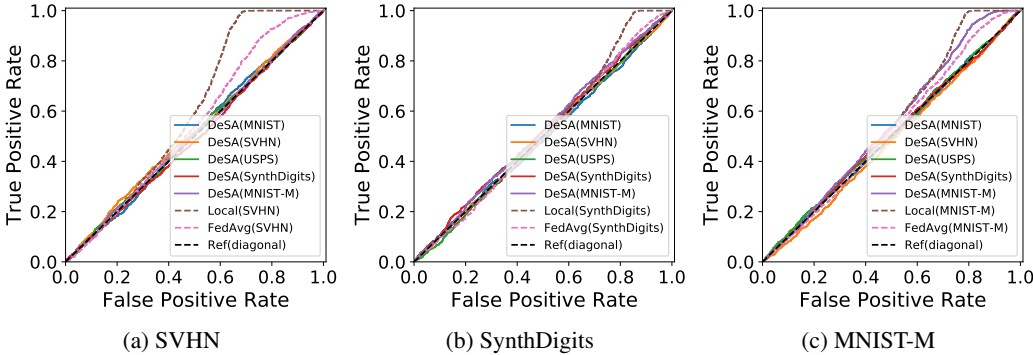

| (a) SVHN | (b) SynthDigits | (c) MNIST-M |

Figure 5: MIA on the models trained by SVHN, SynthDigits, and MNIST-M clients. Observe that the synthetic data sharing of DESA does not reveal other clients' local data identity information.

The goal of the experiment is to investigate whether our local model is vulnerable to MIA, namely leaking information about local real datasets' membership. To compare and demonstrate the effectiveness of the chosen attack, we also present results from local training and FedAvg training. We conduct MIA experiments using DIGITS. The MIA for local training and FedAvg is related to real local training data. Since we use synthetic anchor data generated from other clients with data distillation, we also provide MIA results for inferring real data of other clients. For example, if attacking SVHN's local model, local training and FeAvg report the MIA results on SVHN only, while we also report MIA results on MNIST, USPS, SynthDigits, MNIST-M for DESA.

Using the metric in Carlini et al. (2022a), the results are shown in Figure 5. The Ref(diagonal) line indicates MIA **cannot** tell the differences between training and testing data. If the line bends towards True Positive Rate, it means the membership form the training set can be inferred. It is shown that all the MIA curves of targeted and other cients lie along the Ref line for DESA's model, which indicates that the membership of each training sets is well protected given the applied attack. While the curves for the MIA attacks on FedAvg and local training with SVHN dataset are all offset the Ref (diagonal) line towards True Positive, indicating they are more vulnerable to MIA and leaking training data information.

## C    DIFFERENTIAL PRIVACY FOR DATA SYNTHESIS

To enhance the data privacy-preservation on shared synthetic anchor data, we apply the Differential Privacy stochastic gradient descent (DP-SGD) (Abadi et al., 2016) for the synthetic image generation. DP-SGD protects local data information via noise injection on clipped gradients. In our experiments, we apply Gaussian Mechanism for the inejcted noise. Specifically, we first sample a class-balanced subset from the raw data to train the objective 3. We set up the batch size as 256. For each iteration, we clip the gradient so that its $l_2$-norm is 2. The injected noises are from $\mathcal{N}(0, 1.2)$. This step ensures $(\epsilon, \delta)$-DP with $(\epsilon, \delta)$ values in {(7.622, 0.00015), (10.3605, 0.00021), (8.6677, 0.00017), (7.3174, 0.00014), (7.6221, 0.00015)} guarantees for {MNIST, SVHN, USPS, SynthDigits, MNIST-M}, respectively. We visualize the non-DP and DP synthetic images from each clients in DIGITS in Figure 6 and Figure 7, respectively. One can observe that the synthetic data with DP mechanism is noisy and hard to inspect the individual information of the raw data.

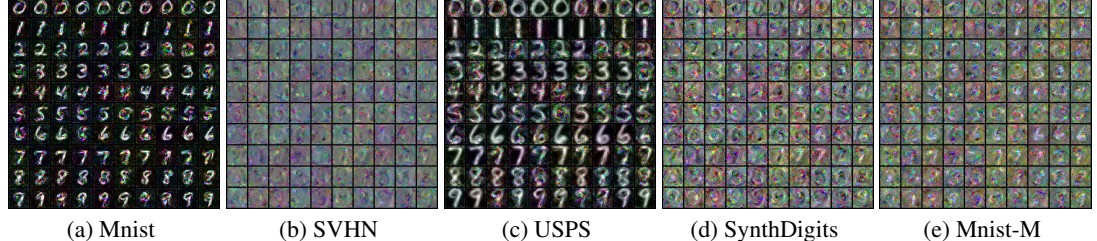

| (a) Mnist | (b) SVHN | (c) USPS | (d) SynthDigits | (e) Mnist-M |
|-----------|----------|----------|-----------------|-------------|

Figure 6: Visualization of the global and local synthetic images from the `DIGITS` dataset. (a) visualized the MNIST client; (b) visualized the SVHN client; (c) visualized the USPS client; (d) visualized the SynthDigits client; (e) visualized the MNIST-M client; (f) visualized the server synthetic data.

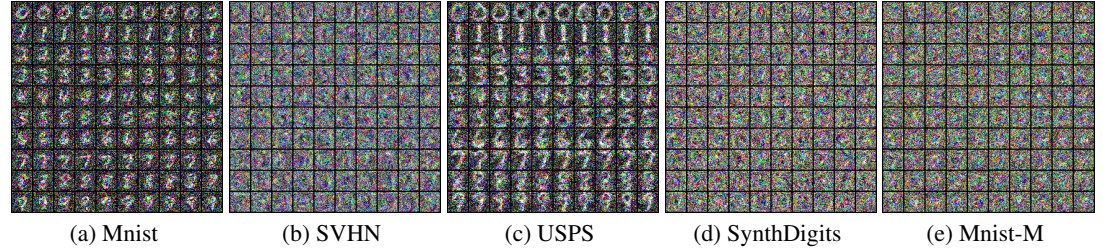

| (a) Mnist | (b) SVHN | (c) USPS | (d) SynthDigits | (e) Mnist-M |
|-----------|----------|----------|-----------------|-------------|

Figure 7: Visualization of the global and local synthetic images from the `DIGITS` dataset with **DP mechanism**. (a) visualized the MNIST client; (b) visualized the SVHN client; (c) visualized the USPS client; (d) visualized the SynthDigits client; (e) visualized the MNIST-M client; (f) visualized the server synthetic data.

We replace the synthetic data by DP synthetic data and perform `DIGITS` experiments, and the result is shown in Table 3. It can be observed that although DESA's performance slightly drops due to the DP mechanism, the averaged inter and intra-accuracy are in the second place, which indicates that DESA is robust as long as we can synthesize images that roughly captures the global data distribution.

Table 3: We add the the results for DESA trained with DP synthetic anchor data into our Table 2. The best result is marked as **bold**, and the second best is marked as blue. The table shows that DESA with DP synthetic anchor data can still has comparable results as DESA with non-DP synthetic data.

|  |  |  | DIGITS | | | | | |
|---|---|---|---|---|---|---|---|---|
|  |  |  | MN(C) | SV(A) | US(C) | Syn(A) | MM(C) | Avg |
| FedHe |  | intra | **98.89** | 83.33 | 98.76 | 93.89 | 94.31 | 93.84 |
|  |  | inter | 49.67 | 62.50 | 37.67 | 70.77 | 65.89 | 57.30 |
| FedDF | C | intra | 92.29 | 20.73 | 86.18 | 15.19 | 51.57 | 53.19 |
|  |  | inter | 34.85 | 12.39 | 18.28 | 17.21 | 37.28 | 24.00 |
|  | F | intra | 92.59 | 19.56 | 93.33 | 77.86 | 69.83 | 70.63 |
|  |  | inter | 39.03 | 18.21 | 30.66 | 54.02 | 54.26 | 39.24 |
| FCCL | C | intra | 11.25 | 19.10 | 13.66 | 10.08 | 11.25 | 13.07 |
|  |  | inter | 13.52 | 11.56 | 12.92 | 13.82 | 13.52 | 13.07 |
|  | F | intra | 95.01 | 72.92 | 94.73 | 86.11 | 90.10 | 87.78 |
|  |  | inter | 34.29 | 58.04 | 29.62 | 57.28 | 60.46 | 47.94 |
| DESA($D_{\text{VHL}}^{Syn}$) |  | intra | 98.69 | 80.09 | 98.66 | 90.68 | 94.40 | 92.50 |
|  |  | inter | 43.33 | 57.51 | 28.26 | 62.02 | 67.69 | 51.76 |
| DESA |  | intra | 98.78 | **86.42** | **98.87** | **95.21** | **94.54** | **94.77** |
|  |  | inter | **65.59** | **80.14** | **64.14** | **78.97** | **69.98** | **71.76** |
| DESA(DP) |  | intra | 98.80 | 86.15 | 98.55 | 94.42 | 94.21 | **94.42** |
|  |  | inter | 62.17 | 74.45 | 55.09 | 76.64 | 69.01 | 67.47 |

# D    COMMUNICATION OVERHEAD

As noted in Section 3.1, DESA only requires sharing logits w.r.t. Global synthetic data during training. Thus it has a relatively low communication overhead compared to baseline methods which require sharing model parameters. For fair comparison, we analyze the communication cost based on the number of parameters Pre-FL and During-FL in Table 4. Note that we show the number of parameters for one communication round for During-FL, and the total communication cost depends on the number of global iterations. One can observe that sharing logits can largely reduce the communication overhead. For example, if we use ConvNet as our model, set IPC=50, and train for 100 global iteration, the total number of parameters for communication for DeSA will be 30.7 K $\times$ 50 (Pre-FL) + 10 (number of classes) $\times$ 50 (images/class) $\times$ 10 (logits/image) $\times$ 100 (global iteration) = 2.04M. In comparison, baseline methods need to share 0 (Pre-FL) + 320K (parameters/iteration) $\times$ 100 (global iteration) = 32M, which is much larger than DeSA. Under model heterogeneity experimental setting, clients using AlexNet would suffer even higher total communication cost, which is 0 (Pre-FL) + 1.87M (parameters/iteration) $\times$ 100 (global iteration) = 187M.

Table 4: Comparison of communication overhead. Note that for DESA, we only share virtual global anchor logits during training. The total communication cost counts the total parameter transferred for 100 global iterations.

|           | ConvNet | AlexNet | Global Anchor Logits |
|-----------|---------|---------|----------------------|
| Pre-FL    | 0       | 0       | 30.7 K $\times$ IPC  |
| During-FL | 320 K   | 1.87 M  | 100 $\times$ IPC     |
| Total     | 32M     | 187M    | 40.7K $\times$ IPC   |

# E    DATASETS AND SYNTHETIC IMAGES

**Detailed Information of Selected Datasets**   1) `DIGITS`={MNIST (LeCun et al., 1998), SVHN (Netzer et al., 2011), USPS (Hull, 1994), SynthDigits (Ganin & Lempitsky, 2015), MNIST-M (Ganin & Lempitsky, 2015)} consists of 5 digit datasets with handwritten, real street and synthetic digit images of $0, 1, \cdots, 9$. Thus, we assume 5 clients for this set of experiments. We use `DIGITS` as baseline to show DESA can handle FL under large domain shift.
2) `OFFICE`={Amazon (Saenko et al., 2010), Caltech (Griffin et al., 2007), DSLR (Saenko et al., 2010), and WebCam (Saenko et al., 2010)} consists of four data sources from Office-31 (Saenko et al., 2010) (Amazon, DSLR, and WebCam) and Caltech-256 (Griffin et al., 2007) (Caltech), resulting in four clients. Each client possesses images that were taken using various camera devices in different real-world environments, each featuring diverse backgrounds. We show DESA can handle FL under large domain shifted `real-world` images using `OFFICE`.
3) `CIFAR10C` consists subsets extracted from Cifar10-C (Hendrycks & Dietterich, 2019), a collection of augmented Cifar10 (Krizhevsky et al., 2009) that applies 19 different corruptions. We employ a Dirichlet distribution with $\beta = 2$ for the purpose of generating three partitions within each distorted non-IID dataset. As a result, we have 57 clients with domain- and label-shifted datasets.

**Synthetic Data Generation**   We fix ConvNet as the backbone for data synthesis to avoid additional domain shift caused by different model architectures. We set learning rate to 1 and use SGD optimizer with momentum = 0.5. The batch size for `DIGITS` and `CIFAR10` is set to 256, while we use 32 for `OFFICE` as it's clients has fewer data points. For the same reason, e use 500 synthetic data points for `DIGITS` and `CIFAR10C`, and we set 100 synthetic data points for `OFFICE`. The training iteration for `DIGITS` and `OFFICE` is 1000, and we set 2000 for `CIFAR10C` since it contains more complex images. The numbers of synthetic data for clients are 500 for {`DIGITS`, `CIFAR10C`} and 100 for {`OFFICE`}, respectively.

We show the local synthetic images and global anchor images of `DIGITS`, `OFFICE`, and `CIFAR10C` in Figure 8, Figure 9, and Figure 10, respectively.

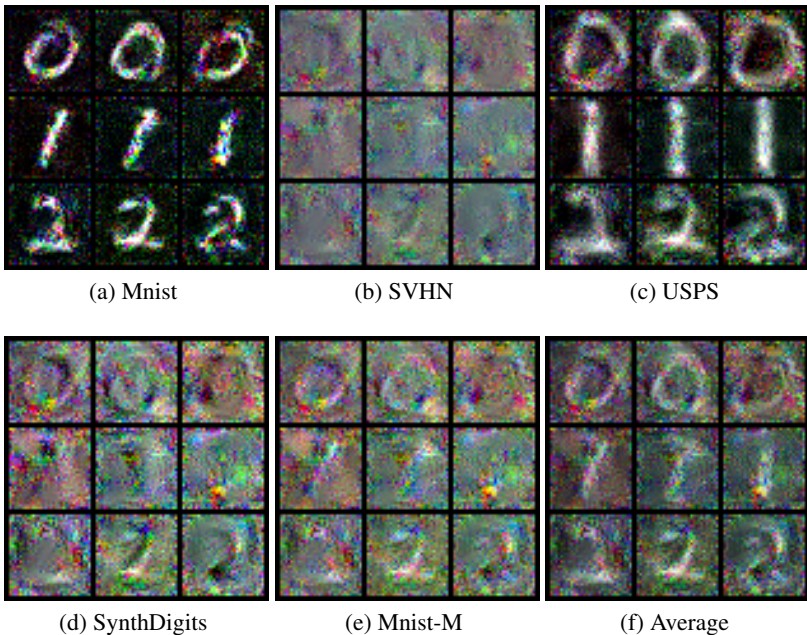

Figure 8: Visualization of the sampled global and local synthetic images from the `DIGITS` dataset. (a) visualized the MNIST client; (b) visualized the SVHN client; (c) visualized the USPS client; (d) visualized the SynthDigits client; (e) visualized the MNIST-M client; (f) visualized the server synthetic data.

## F  MODEL ARCHITECTURES

We use ConvNet to perform data distillation for the best synthesis quality. For model hetero genenity scenarios, we randomly select classification model architectures from {AlexNet, ConvNet}.

Table 5: AlexNet architecture. For the convolutional layer (Conv2D), we list parameters with a sequence of input and output dimensions, kernel size, stride, and padding. For the max pooling layer (MaxPool2D), we list kernel and stride. For a fully connected layer (FC), we list input and output dimensions.

| Layer | Details |
|-------|---------|
| 1 | Conv2D(3, 128, 5, 1, 4), ReLU, MaxPoo2D(2, 2) |
| 2 | Conv2D(128, 192, 5, 1, 2), ReLU, MaxPoo2D(2, 2) |
| 3 | Conv2D(192, 256, 3, 1, 1), ReLU |
| 4 | Conv2D(256, 192, 3, 1, 1), ReLU |
| 5 | Conv2D(192, 192, 3, 1, 1), ReLU, MaxPoo2D(2, 2) |
| 22 | FC(3072, num_class) |

## G  MORE RELATED WORK

### G.1  MODEL HOMOGENEOUS FEDERATED LEARNING

We list down different Model Homogeneous FL approaches in decentralized FL and collaborative methods that are relevant to our setting.

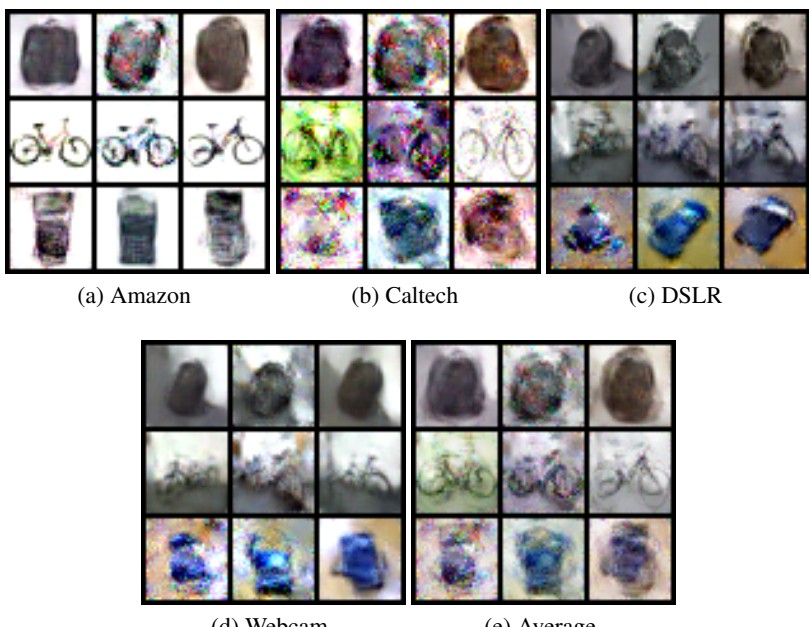

(a) Amazon        (b) Caltech        (c) DSLR

(d) Webcam        (e) Average

Figure 9: Visualization of the sampled global and local synthetic images from the OFFICE dataset. (a) visualized the Amazon client; (b) visualized the Caltech client; (c) visualized the DSLR client; (d) visualized the Webcam client; (e) visualized the averaged synthetic data.

Table 6: ConvNet architecture. For the convolutional layer (Conv2D), we list parameters with a sequence of input and output dimensions, kernel size, stride, and padding. For the max pooling layer (MaxPool2D), we list kernel and stride. For a fully connected layer (FC), we list the input and output dimensions. For the GroupNormalization layer (GN), we list the channel dimension.

| Layer | Details |
|-------|---------|
| 1 | Conv2D(3, 128, 3, 1, 1), GN(128), ReLU, AvgPool2d(2,2,0) |
| 2 | Conv2D(128, 118, 3, 1, 1), GN(128), ReLU, AvgPool2d(2,2,0) |
| 3 | Conv2D(128, 128, 3, 1, 1), GN(128), ReLU, AvgPool2d(2,2,0) |
| 4 | FC(1152, num_class) |

### G.1.1 DECENTRALIZED FEDERATED LEARNING

In order to tackle training a global model without a server, Decentralized FL methods communicate a set of models through diverse decentralized client-network topologies (such as a ring - (Chang et al., 2018), Mesh - (Roy et al., 2019), or a sequential line (Assran et al., 2019)) using different communication protocols such as Single-peer(gossip) or Multiple-Peer(Broadcast). (Yuan et al., 2023a; Sheller et al., 2019; 2020) pass a single model from client to client similar to an Incremental Learning setup. In this continual setting, only a **single model** is trained. (Pappas et al., 2021; Roy et al., 2019; Assran et al., 2019) pass models and aggregate their weights similar to conventional FL. Since these models use averaged aggregation techniques similar to FedAvg, most of these methods assume client **model homogeneity**. DESA's client network topology is similar to that of a Mesh using the broadcast-gossip protocol, where every client samples certain neighbours in each communication round for sharing logits.

None of the works above aim to train various client model types without a server, which is our goal.

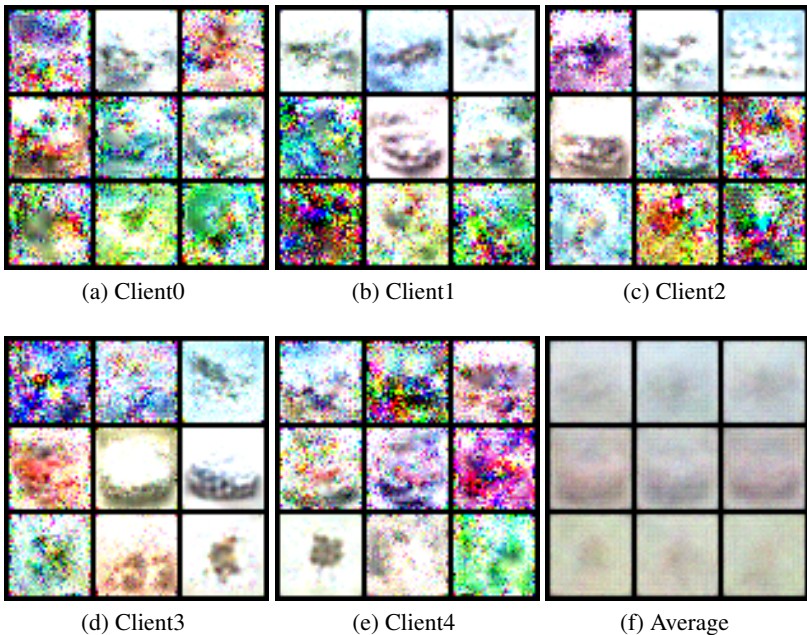

(a) Client0      (b) Client1      (c) Client2

(d) Client3      (e) Client4      (f) Average

Figure 10: Visualization of the sampled global and local synthetic images from the first 5 clients in `CIFAR10C` dataset. (a) visualized the first client; (b) visualized the second client; (c) visualized the third client; (d) visualized the forth client; (e) visualized the fifth client; (f) visualized the server synthetic data.

### G.1.2 COLLABORATIVE METHODS

Fallah et al. (2020) uses an MAML(model agnostic meta learning) framework to explicitly train model homogeneous client models to personalize well. The objective function of MAML evaluates the personalized performance assuming a one-step gradient descent update on the subsequent task. (Huang et al., 2021b) modifies the personalized objective by adding an attention inducing term to the objective function which promotes collaboration between pairs of clients that have similar data.

Ghosh et al. (2022) captures settings where different groups of users have their own objectives (learning tasks) but by aggregating their private data with others in the same cluster (same learning task), they can leverage the strength in numbers in order to perform more efficient personalized federated learning (Donahue & Kleinberg, 2021) uses game theory to analyze whether a client should jointly train with other clients in a conventional FL setup [2.1] assuming it's primary objective is to minimize the MSE loss on its own private dataset. They also find techniques where it is more beneficial for the clients to create coalitions and train one global model.

All the above works either slightly change the intra-client objective to enable some collaboration between model-homogeneous clients or explicitly create client clusters to collaboratively learn from each other. They do not tackle the general objective function that we do- 2

### G.2 MODEL HETEROGENEOUS FEDERATED LEARNING

Model heterogeneous FL approaches relevant to DESA broadly come under the following two types.

### G.2.1 KNOWLEDGE DISTILLATION METHODS

Gong et al. (2022) proposes FedKD that is a one-shot centralized Knowledge distillation approach on unlabelled public data after the local training stage in-order to mitigate the accuracy drop due to the label shift amongst clients. DENSE (Zhang et al., 2022a) propose one-shot federated learning to generate decision boundary-aware synthetic data and train the global model on the server side. FedFTG (Zhang et al., 2022b) finetunes the global model by knowledge distillation with hard sample

mining. Yang et al. (2021) introduces a method called Personalized Federated Mutual Learning (PFML), which leverages the non-IID properties to create customized models for individual parties. PFML incorporates mutual learning into the local update process within each party, enhancing both the global model and personalized local models. Furthermore, mutual distillation is employed to expedite convergence. The method assumes homogeneity of models for global server aggregation. However, all the above methods are centralized.

### G.2.2 MUTUAL LEARNING METHODS

Papers in this area predominantly use ideas from deep-mutual learning (Zhang et al., 2018) Matsuda et al. (2022) uses deep mutual learning to train heterogeneous local models for the sole purpose of personalization. The method creates clusters of clients whose local models have similar outputs. Clients within a cluster exchange their local models in-order to tackle label shift amongst the data points. However, the method is centralized and each client maintains two copies of models, one which is personalized and one that is exchanged. Li et al. (2021a) has a similar setting to Chan & Ngai (2021), but instead solves the problem in a peer to peer decentralized manner using soft logit predictions on the local data of a client itself. It makes its own baselines that assume model homogeneity amongst clients, also their technique assumes that there is no covariate shift because it only uses local data for the soft predictions. However, their technique can be modified for model heterogeneity. They report personalization(Intra) accuracies only.

### G.3 DATASET DISTILLATION

Data distillation methods aim to create concise data summaries $D_{syn}$ that can effectively substitute the original dataset $D$ in tasks such as model training, inference, and architecture search. Moreover, recent studies have justified that data distillation also preserves privacy (Dong et al., 2022; Carlini et al., 2022b) which is critical in federated learning. In practice, dataset distillation is used in healthcare for medical data sharing for privacy protection (Li et al., 2022). We briefly mention two types of Distillation works below.

### G.3.1 GRADIENT AND TRAJECTORY MATCHING TECHNIQUES

Gradient Matching (Zhao et al., 2020) is proposed to make the deep neural network produce similar gradients for both the terse synthetic images and the original large-scale dataset. The objective function involves matching the gradients of the loss w.r.t weights(parameters) evaluated on both $D$ and $D_{syn}$ at successive parameter values during the optimization on the original dataset $D$. Usually the cosine distance is used to measure the difference in gradient direction. Other works in this area modify the objective function slightly, by either adding class contrastive signals for better stability Lee et al. (2022) or by adding same image-augmentations(such as crop, rotate to both $D$ and $D_{syn}$)(Zhao & Bilen, 2021). A similar technique is that of (Cazenavette et al., 2022) which tries to match the intermediate parameters in the optimization trajectory of both $D$ and $D_{Syn}$. It is very *computationally expensive* because of a gradient unrolling in the optimization. TESLA (Cui et al., 2023) attempts at using linear-algebraic manipulations to give better computational guarantees for Trajectory matching

### G.3.2 DISTRIBUTION MATCHING TECHNIQUES

Distribution matching (Zhao & Bilen, 2023) solves the distillation task via a single-level optimization, leading to a *vastly improved scalability*. More specifically, instead of matching the quality of models on $D$ vs. $D_{syn}$, distribution-matching techniques directly match the distribution of $D$ vs. $D_{syn}$ in a latent encoded space. See 3 for the objective function. CAFE (Wang et al., 2022) further refines the distribution-matching idea by solving a bilevel optimization problem for jointly optimizing a single encoder and the data summary, rather than using a pre-determined set of encoders Adversarial techniques using Distribution matching such as IT-GAN (Zhao & Bilen, 2022) and GAN (Goodfellow et al., 2014) aren't suitable for a serverless setting. Since we aim to mitigate drifts in client-distribution across using our synthetic data, Distribution Matching is a more natural option for our work.

