# OpenReview forum: "Overcoming Data and Model heterogeneities in Decentralized Federated Learning via Synthetic Anchors"
_ICLR.cc/2024/Conference — Submitted to ICLR 2024_

### Official Review · Reviewer_phME · 2023-10-23

**Soundness:** 3 good
**Presentation:** 2 fair
**Contribution:** 2 fair
**Rating:** 3
**Confidence:** 5

**Summary:**

This paper studied the decentralized federated learning with both data and model heterogeneity. To solve this problem, the authors introduced a novel DESA method, which generated global synthetic anchors to guide the local model training. For each client, in addition to standard supervised classification loss, it would also consider the classification loss over synthetic anchors and cross-client knowledge distillation losses for improving the model's generalization performance. Experimental results validated the effectiveness of DESA over baselines with respect to both inter- and intra-client prediction performance.

**Strengths:**

**(1) Originality:** This paper handled both data and model heterogeneity in decentralized federated learning problems without public data. Technically, it proposed to generate synthetic anchor data from each client. Besides, supervised contrastive loss between client data and synthetic data was introduced to mitigate the data heterogeneity among clients. The knowledge distillation loss over synthetic data was designed to mitigate the model heterogeneity among clients. The generalization performance of the proposed DESA method was theoretically analyzed.

**(2) Quality:** DESA used synthetic anchors to solve the issues of decentralized federated learning without public data. It further leveraged contrastive loss and knowledge distillation loss to handle data and model heterogeneity. The hyperparameter analysis in the experiments also validated the impact of those two losses on the proposed DESA method.

**(3) Clarity:** Overall, the presentation of this paper is easy to follow. This paper illustrated the three crucial components of DESA in different subsections. The effectiveness of DESA was evaluated on a variety of benchmarks, including both heterogeneous and homogeneous model settings.

**(4) Significance:** The studied decentralized federated learning is practical in real-world applications, especially when no public data is available among local clients. Thus, the developed method without leveraging public data in this paper can be applied to more general FL problems compared to previous works relying on public data.

**Weaknesses:**

The weaknesses of this paper are summarized below.

(1) The research question is not well motivated. This paper studied decentralized federated learning regarding the performance of every client model on other clients. Traditional FL settings might focus only on the performance of every client model on its own client domain. Thus, it might be more convincing to provide some practical examples to illustrate why inter-client test accuracy should be emphasized in real FL scenarios.

(2) The introduction shows that the proposed approach aims to generate minimal synthetic anchor data to enhance client-model generalization. However, this "minimal" property of generated data is not discussed in the experiments. The ablation study in subsection 5.4 shows that the size of synthetic data can significantly affect the inter-accuracy. Thus, there might exist a trade-off between the model performance and the size of synthetic data. More explanations can be provided here.

(3)  Subsection 3.1 shows that synthetic anchor data is shared amongst the client’s neighbors. However, it is unclear how the neighbor information is defined in the experiments. In addition, it seems that the synthetic anchor data $D^{Syn}$ in Subsection 3.2 simply combines all the anchors $D^{Syn}_i$ within each client.

(4) The definition of distribution matching in Eq. (3) is confusing. First, it is unclear why this term can guarantee the class-imbalanced anchors. How is the function $\psi^{rand}(x|y)$ affected by the class labels? Second, it is defined over all clients $i=1,\cdots, N$. Then why does the generation of anchors within client $i$ rely on the data on other clients? Third, it is not explained whether the minimization of MMD between true data and anchor data would increase the risk of privacy leakage. That is, when anchor data becomes more similar to the true data, it is more likely to include the private domain information.

(5) In the derived generalization in Theorem 1, it assumes (i)  real labeling and synthetic data labeling are similar, and (ii) real labeling and distillation data labeling are also similar. It is confusing how both assumptions can always be guaranteed in real scenarios.

(6) The experimental settings show that for heterogeneous model experiments, multiple baselines are compared, including FedMD, FedDF, FCCL, FedGen, and VHL. But Table 2 only lists the results of FedHe, FedDF, and FCCL.

**Questions:**

(1) The client structure information is not provided in the experiments. Does it imply that all the clients are connected with each other in all experiments?

(2) In subsection 3.3., "$P(\cdot)$ index class category" is confusing. Where is $P(\cdot)$ used in this section?

(3) The communication costs of DESA can be analyzed, because it might include additional anchor sharing and logits sharing compared to baselines.

(4) Some notations used in Theorem 1 are undefined, e.g., $d_{H\Delta H}$, $\lambda(P_i)$, etc. In addition, what does "if $\psi_i \circ P^{Syn} \to \psi_i \circ P^T$ for any $\psi_i$" imply?

(5) Below Proposition 2, it is shown that when the local data heterogeneity is severe, the model learning should rely more on the centralized data, e.g., synthetic data and the extended KD data. This can be verified in the experiments, e.g., how the hyperparameters $\lambda_{REG}$ and $\lambda_{KD}$ can be changed with respect to the data heterogeneity.

(6) "FedSAB" in subsection 5.3 is undefined.

---

> ### Author Response · Authors · 2023-11-20
> **Justification of our motivation for federated mutual learning**
>
> We thank the reviewer for raising this concern. Let us think about an essential scenario: In 2020, hospitals from different regions want to collaboratively train a COVID-19 classification model. Due to quarantine, people cannot travel outside the local area, so the local hospital can only collect local cases with local strains of Covid viruses. If they train a model that only cares about intra-domain, this *personalized* model will suffer from a performance drop when the quarantine restriction is revoked and people from one region (with its local virus strains) will travel to other regions and use other hospitals’ models. Thus, we consider *intra* and *inter*-domain performances are both  important, and allowing each client model to generalize well to other client domains can potentially improve model robustness. Thus, In this work, we consider a different FL setting without sever and define it as  *decentralized federated mutual learning*, which shares the same motivation as the existing work [1].
>
> [1] Huang W, Ye M, Du B. Learn from others and be yourself in heterogeneous federated learning. InProceedings of the IEEE/CVF Conference on Computer Vision and Pattern Recognition 2022 (pp. 10143-10153).

---

> ### Author Response · Authors · 2023-11-20
> **The effect of IPC**
>
> Finding the minimal amount synthetic data is not the focus of this work. We have replaced “minimal” with “small” in our revision for clarification. Following your suggestion, we have performed a more detailed hyperparameter search on IPC. Specifically, we search IPC from {5, 10, 20, 50, 100, 200} and the experimental results are updated in our revision.
> Overall, blindly increasing the IPC does not guarantee to obtain optimal intra- and inter-accuracy. It will cause the loss function to be dominated by the last 2 terms of Eq. 8, \textit{i.e.,} by synthetic data rather than minimizing the empirical risk of classification on cross-entropy loss. However, synthesizing larger number of synthetic data may degrade its quality, and the sampled batch for $\mathcal{L}_{REG}$ may fail to capture the distribution.

---

> ### Author Response · Authors · 2023-11-20
> **Explanation of the global synthetic data and the neighbor information**
>
> Regarding how we merge D^{syn}: We perform simple interpolation (averaging) among clients as it is shown that using this mixup strategy can improve model fairness [1].
>
> Regarding how we define our neighbors: We assume the client nodes form a complete graph in DIGITS and OFFICE experiments. For CIFAR10C experiments, we randomly sample neighboring clients for each round.
> We have added the clarification in our revision.
>
>
> [1]  Chuang CY, Mroueh Y. Fair Mixup: Fairness via Interpolation. InInternational Conference on Learning Representations 2020 Oct 2.

---

> ### Author Response · Authors · 2023-11-20
> **Clarification of Eq. 3 and synthetic data generation**
>
> We thank the reviewer for the careful review and clarification questions.
>
> **Typo on notation:** We apologize for the typo. There shouldn’t be a summation over i to N, so the client i’s data is not affected by client j, $\forall i \neq j$. We thank the reviewer for the careful review and have corrected it in our revision, and we sincerely apologize for causing your confusion.
>
> **Class-balanced guarantee:** We synthesize data based on each class, i.e., for synthesizing data for class c, we will sample real data from P(x|y=c). As a result, we can control the number of synthetic data for each class c, and generate a class-balanced synthetic data set.
>
> **Potential privacy leakage:** As discussed in [2], using MMD distilled data to train a model can defend against reconstruction attacks and membership inference attacks, which is empirically justified in our Appendix B. For further privacy preservation, we also show we can apply Differential Privacy when distilling synthetic data in Appendix C.
>
> [1] Bo Zhao and Hakan Bilen. Dataset condensation with distribution matching. In Proceedings of the IEEE/CVF Winter Conference on Applications of Computer Vision, pp. 6514–6523, 2023.
>
> [2] Dong T, Zhao B, Lyu L. Privacy for free: How does dataset condensation help privacy?. InInternational Conference on Machine Learning 2022 Jun 28 (pp. 5378-5396). PMLR.

---

> ### Author Response · Authors · 2023-11-20
> **Justification and explanation of the labelling function for synthetic data**
>
> We thank the reviewer for his astute observation about the labelling functions.The only mathematical assumption that the theorem makes is that the encoded distributions of the source $P_s$ and the global data domain $P_t$ converge to each other. Practically, this is enforced by the anchor term in the loss function, $\mathcal{L}_{REG}$.
>
> Coming to the interpretation of the bound through the similarity of the labeling functions.
>
> Firstly, it is important to note that the theorem does not require both of the error terms to be small. As long as any single one of the terms is low, we can change the component weights $\alpha$ to rely more on that term , so that we get a tighter generalization guarantee. These component weights alpha are directly tied to the regularization weights $\lambda_{REG}$ and $\lambda_{KD}$.
> The explanation for either of the labelling functions being close to $f_T$ is given below,
>
> 1. The real data and the synthetic data will have similar labeling functions because their distributions are enforced to be similar under an MMD metric during the data generation process. - (Eq. 3) . Therefore $f^{syn}$ will be close to $f_T$
>
> 2. The statement about the extended KD data is a bit more subtle. It is crucial to note that $D_{KD}^{syn}$ depends on the client models $M_j$  and **is not a static distribution.**  Towards the end of training, when the empirical KD loss consistently decreases and generalized models are learned, the labeling function of the extended KD data $D_{KD}^{syn}$, which represents the consensus knowledge of all the domains,  will correctly classify any example drawn from the global data  domain  $P_T$.
> Therefore, we project that the labeling function $f_{KD}^{syn}$ will be close to $f_T$.

---

> ### Author Response · Authors · 2023-11-20
> **Explanation of the method selections for model heterogeneous experiments**
>
> The comparison results with FedGen are presented in Homogeneous Model Experiments (Section 5.3). We did not include FedGen in Table 2 as it is not applicable to model heterogeneous scenarios discussed in table 2. FedMD[1] was not considered to be included due to its poor performance, consistent with the observation report in its similar method FedHe [3]. We have corrected the paragraph by removing FedGen and FedMD and adding FedHe in our revision.
>
> [1] Li D, Wang J. Fedmd: Heterogenous federated learning via model distillation. arXiv preprint arXiv:1910.03581. 2019 Oct 8.
>
> [2] Zhu Z, Hong J, Zhou J. Data-free knowledge distillation for heterogeneous federated learning. InInternational conference on machine learning 2021 Jul 1 (pp. 12878-12889). PMLR.
>
> [3] Chan YH, Ngai EC. Fedhe: Heterogeneous models and communication-efficient federated learning. In2021 17th International Conference on Mobility, Sensing and Networking (MSN) 2021 Dec 13 (pp. 207-214). IEEE.

---

> ### Author Response · Authors · 2023-11-20
> **Clarification on the client structure**
>
> We assume the client nodes form a complete graph in DIGITS and OFFICE experiments. For CIFAR10C experiments, we randomly sample neighboring clients for each round.

---

> ### Author Response · Authors · 2023-11-20
> **Clarification on $P(\cdot)$**
>
> We thank the reviewer for the careful review. The $P(\cdot)$ is unused and we have removed $P(\cdot)$ in our revision.

---

> ### Author Response · Authors · 2023-11-20
> **Clarification on the communication cost of DeSA**
>
> We thank the reviewer for the suggestion. As noted in our Section 3.1, DeSA only requires sharing logits w.r.t. Global synthetic data during model training. Thus it has a relatively low communication overhead compared to baseline methods which require sharing model parameters. For example, if we use ConvNet as our model and set IPC=50, the number of parameters for communication for DeSA will be 10(number of classes) x 50(images/class) x 10(logits/image) = 5k. In comparison, baseline methods need to share 320K parameters (the number of parameters of ConvNet model), which is much larger than DeSA. We also add the discussion in Appendix D in our revision.

---

> ### Author Response · Authors · 2023-11-20
> **Explanation of the notations in Theorem 1**
>
> We thank the reviewer for pointing this out.
>
> The exact mathematical definitions have been added to our revision.
> Below, we explain their meaning in words.
>
> 1. $d_{H \Delta H}$ is a distance metric that relates to the maximum difference in probability assigned to the space of disagreements between any two hypothesis functions drawn from H.
> 2. $\lambda(P_i)$ refers to the combined generalization loss of the best joint model on both local and client domains.
> 3. The condition implies that there should exist a function $\psi$  such that the encoded distributions of $P_s$ and $P_t$  converge to each other in probability.
> This convergence condition is enforced by the loss $\mathcal{L_reg}$, which acts as an anchor to pull all encoded distributions together.

---

> ### Author Response · Authors · 2023-11-20
> **Clarification of Proposition 2**
>
> We clarify that the statement “when the local data heterogeneity is severe, the model learning should rely more on the centralized data” is on the assumption  that “ synthetic and the extended KD datasets are similar to the global ones” Thus the LHS of Eq(9) will be dominated by $ \epsilon_{{P}^T}(f^{Syn})$ and $ \epsilon_{{P}^T}(f_{\text{KD}}^{Syn})$. It is intuitive that if synthetic data is ideal aligned with global distribution and we fully rely on synthetic data in local training, the performance gain between training on synthetic data vs on local data is expected to be larger when local data is more heterogeneous. Also, as noted in Theorem 1, the performance is not solely related to hyperparameters $\lambda_{REG}$ and $\lambda_{KD}$. It will be also affected (or dominated) by the fourth term of Eq 8, when we rely on sufficient number of ideal synthetic data (aligned with global distribution).

---

> ### Author Response · Authors · 2023-11-20
> **FedSAB in Subsection 5.3**
>
> Sorry for the typo, we have corrected it in our revision.

---

> > ### Comment · Reviewer_phME · 2023-11-20
> > **Comments**
> >
> > Thanks for the rebuttals. But I have other follow-up concerns.
> >
> > (1) It shows the impact of IPC on the proposed DESA framework.
> > - It affects the trade-off between inter- and intra-performance. In the context of decentralized federated mutual learning, how should the optimal trade-off between inter- and intra-performance be defined? For example, in COVID-19 classification, larger IPC can lead to improved inter-accuracy but degraded intra-accuracy. In this case, how will IPC be selected?
> > - The results of Figure 4(c) are also affected by the hyperparameters $\lambda_{KD}$ and $\lambda_{REG}$. Does Figure 4(c) consider the same hyperparameters for IPC?
> >
> > (2) The explanation of the labeling function is still unclear to me.
> > - What does the assumption of "the encoded distributions of the source $P_s$ and the global data domain $P_t$ converge to each other" mean? What are "encoded distributions of the source" here? How can this be enforced by $\mathcal{L}_{REG}$?
> > - How can the MMD in Eq. (3) guarantee that $f^{syn}$ is similar to $f_T$?
> >
> > (3) The computation of the communication cost of DeSA is confusing. It is shown that DESA only requires sharing logits. But all the three terms in Eq. (7) involve the global synthetic data $D^{Syn}$. Would $D^{Syn}$ be shared in FL?
> >
> > (4) The experiments consider only synthetic structure information on common image data sets. It might be more convincing to include some real-world data sets (e.g., COVID-19 classification) which indicate both inter- and intra-accuracy should be emphasized. This can also verify the impact of structure information on the proposed DeSA method in real scenarios.

---

> > > ### Author Response · Authors · 2023-11-21
> > > **Clarification of the impact of IPC on DeSA**
> > >
> > > > It affects the trade-off between inter- and intra-performance. In the context of decentralized federated mutual learning, how should the optimal trade-off between inter- and intra-performance be defined?
> > >
> > > The definition of the “optimal trade-off” may vary case-by-case and depends on the clients. Here, we provide an intuitive selection - intra- and inter- accuracy are equally important under the consideration of federated mutual learning. Thus, in the ideal case, we would suggest searching for the hyperparameter that results in the performance on the tip-right corner of intra- and inter-accuracy plot. For example, in our Figure 4(c), we tend to choose IPC=20 as it has the best balance between intra- and inter-accuracy.
> > >
> > > > For example, in COVID-19 classification, larger IPC can lead to improved inter-accuracy but degraded intra-accuracy. In this case, how will IPC be selected?
> > >
> > > Thank you for raising the point about the impact of IPC on inter- and intra-accuracy in COVID-19 classification. We note the reviewer's statement ““For example, in COVID-19 classification, larger IPC can lead to improved inter-accuracy but degraded intra-accuracy”. However, our study did not specifically investigate this scenario, and thus we did not conduct experiments related to COVID-19 classification. As such, we are unable to confirm or discuss the effects of varying IPC in this particular case. We would appreciate further clarification or references regarding this example to better understand the context and implications of the statement.
> > >
> > > In the results showcased in Figure 4(c), we observe an increase in inter-accuracy and a decrease in intra-accuracy as IPC changes from 20 to 50. This increase in inter-accuracy is consistent with our Theorem 1. The observed decrease in intra-accuracy can be attributed to the inclusion of more synthetic data in local training, leading to a tendency for model performance to align with the global distribution. Nevertheless, when the global synthetic data is not optimal, this alignment adversely affects local performance. We believe these observations are influenced not only by IPC but also by the quality of the synthetic data.
> > >
> > > In practical scenarios, selecting IPC is a heuristic process influenced by various factors, including the nature of the dataset and the application, the desired emphasize on inter- and intra-accuracy, the device's data storage capacity, computational limitations, and the communication cost associated with transmitting synthetic data (one-time) and logits. This multifaceted approach ensures that IPC selection is tailored to specific use-case requirements and constraints.
> > >
> > > > The results of Figure 4(c) are also affected by the hyperparameters lambda_{KD} and lambda_{REG}. Does Figure 4(c) consider the same hyperparameters for IPC?
> > >
> > > We use the same hyperparamters when varying IPC (both $\lambda_{KD}$ and $\lambda_{REG}$ equal to 1, as stated in the experimental setup). The observation aligns our theoretical analysis with the fixed $\lambda's$.

---

> > > ### Author Response · Authors · 2023-11-21
> > > **Explanation of the communication cost of DeSA**
> > >
> > > We consider the communication overhead of the information transferred between clients when collaboratively training a model. Thus, we calculate the communication cost *during FL training* and only consider the logits of global synthetic data. We would like to note that we share the synthetic data only once before the FL starts, and the clients don't have to update/receive global synthetic data during FL.

---

> > > ### Author Response · Authors · 2023-11-21
> > > **Thank you for the follow-up comments**
> > >
> > > We thank the reviewer for the follow up comments and have tried our best to solve your doubts and confusion. We are happy to answer any new question that may arise. Meanwhile, we sincerely hope the reviewer can reconsider the rating of DeSA.
> > >
> > > Best Regards,
> > >
> > > DeSA authors

---

> > > > ### Comment · Reviewer_phME · 2023-11-21
> > > > **Comments**
> > > >
> > > > (1) Regarding the assumption on $P^{Syn}$ and $P^T$, it may not hold that they are similar when true client distribution is class-imbalanced. In this case, $P^{Syn}$ would always be given by class-balanced synthetic samples. Then it is not reasonable to assume that $P^{Syn}$ and $P^T$ are similar distributions (balanced vs. imbalanced distributions).
> > > >
> > > > (2) More theoretical analysis on the similarity between $f^{Syn}$ and $f^T$ will be more convincing. It directly affects the tightness of the generalization error bound in Theorem 1. For example, MMD corresponds to the empirical distribution similarity of synthetic samples and real samples. But labeling functions in [Ben-David S, 2010] are defined over the expected functions.
> > > >
> > > > (3) For a fair comparison, it is better to consider all the involved communication costs in a FL method, including pre-processing and online information sharing during training.

---

> > > > > ### Author Response · Authors · 2023-11-22
> > > > > **Thank you for the follow-up comments and the active discussion**
> > > > >
> > > > > > Clarifying Misunderstanding on Theorem and Assumption
> > > > >
> > > > > We appreciate the opportunity to clarify your potential misunderstanding regarding our theorem and assumptions. It's important to note that **our approach does NOT require the the $P^{Syn}$ and $P^T$ to be similar. Instead, we require the embeddings' distribution $\psi \circ P^{Syn}$ and $\psi \circ P^T$ to be similar**. In our algorithm, we achieve this by adding the $L_{\rm reg}$ term, which is a per-class similarity loss. By applying this method, we believe that our approach effectively addresses issues arising in class-imbalanced scenarios.
> > > > >
> > > > > > Addressing Concerns on Generalization Error
> > > > >
> > > > > Regarding your concern that similarity between $f^{Syn}$ and $f^T$ will directly affects the generalization error mentioned in Theorem 1, we want to clarify that 1) In our Theorem 1, the error term w.r.t. $f^{Syn}$ is a generally satisfied for all possible synthetic labeling function. 2) Although the MMD is measuring on the empirical distribution, following Lemma 1&5 in [1], it can be proved that the bound of $f^{Syn}$ will only have an error with order $O(\sqrt{1/N})$ between the empirical and expected versions, where $N$ is the empirical dataset size.
> > > > >
> > > > > > Fair comparison of communication overhead
> > > > >
> > > > > We thank the reviewer for the suggestion. In our revised Appendix, we added the total communication cost, and updated the table as follows:
> > > > > |               |ConvNet|AlexNet|Global Anchor Logits|
> > > > > |----------|----------|----------|----------|
> > > > > |Pre-FL.         |0.           |0.           |30.7K$\times$IPC|
> > > > > |During-FL.   | 320K    |1.87M   |100 $\times$ IPC|
> > > > > |Pre-FL.         |32M      |187M   |40.7K $\times$ IPC|
> > > > >
> > > > > We appreciate the reviewer's feedback and the chance to clarify and improve DeSA, and it's our pleasure to answer further questions if there is any.
> > > > >
> > > > > [1] Ben-David S, Blitzer J, Crammer K, Kulesza A, Pereira F, Vaughan JW. A theory of learning from different domains. Machine learning. 2010 May;79:151-75.

---

> ### Author Response · Authors · 2023-11-21
> **Explanation of the labeling function**
>
> > What does the assumption of "the encoded distributions of the source $P_s$ and the global data domain $P_t$ converge to each other" mean?
>
> In our paper, we represent the $i$th client model as two parts: the feature encoder $\psi_i$ and classification head $\rho_i$. $\psi_i$ will map the raw data (i.e. the image data in our experiments) to the embedding representation, and $\rho_i$ will output the prediction on top of the embeddings. In our theoretical analysis, the core assumption is that the distributions of embedding representations under the synthetic dataset $P^{Syn}$ and global dataset $P^T$ across clients are similar. We express this mathematically as $\psi_i \circ P^{Syn} \rightarrow \psi_i \circ P^{T}$, where $\psi_i \circ P^{Syn}$ is the distribution of the embeddings over the synthetic dataset and $\psi_i \circ P^{T}$ is that over the global dataset.
>
> > What are "encoded distributions of the source" here?
>
> The encoded distributions represent the embeddings derived from the global synthetic dataset. In this context, we use the term 'source data domain' to specifically refer to the synthetic dataset of all the clients.
>
> > How can this be enforced by $\mathcal{L}_{REG}$
>
> If we go back to the definition of $L_{REG}$ (i.e., Eq (4) in the paper), it minimizes the per-class distances of the embeddings over the synthetic dataset and the true datasets. When it is approaching zero, it means that the per-class averaged embeddings over the synthetic and true datasets are the same. When we minimize $\mathcal{L}_{REG}$ for every client model, the synthetic data behaves as an anchor, pulling each domain’s encoded representation close to the encoded representation of the synthetic data.
> Therefore, as the models learn $\psi_i$, they will learn to project the synthetic data distribution $P_s$ into a domain invariant representation $\psi_i \circ P_T$
>
> > How can the MMD in Eq. (3) guarantee that $f^{syn}$ is similar to $f_T$?
>
> 1. The MMD in Eq. (3) is a label-conditional distance. It requires that the generated local synthetic data and local client data behave similarly under each class.
>
> 2. The global synthetic data $D^{syn}$ is the aggregation of all the local synthetic data. Since eq(3) guarantees that the local synthetic data distribution is similar to the local client data distribution, the global synthetic data distribution becomes representative of $P_T$
>
> 3. The labelling function from [1] is characteristic to the distribution it is defined upon, therefore, the labelling function $f_{syn}$ is close to $f_T$.
>
> [1] Ben-David S, Blitzer J, Crammer K, Kulesza A, Pereira F, Vaughan JW. A theory of learning from different domains. Machine learning. 2010 May;79:151-75.

---

> ### Author Response · Authors · 2023-11-21
> **Clarification of DeSA and its experiments**
>
> We wish to clarify that for this study, **our choice of datasets was guided by the objective to align with the widely recognized standards** in dataset distillation literature [1,2,3]. The selected datasets, commonly used in FL research, enable direct comparisons with existing methods and model both feature and label heterogeneities, ensuring our findings are relevant and interpretable in current research contexts.
>
> We acknowledge that the distribution-based distillation method employed in our study is optimized for the datasets we selected and may not be directly applicable to high-resolution images, which often require more sophisticated distillation strategies, such as [1,2].  Due to time constraints and the specialized nature of medical image processing for COVID19 classification task, it wasn't feasible to include these in this rebuttal phase.
>
> Notably, **obtaining good distilled datasets can be a standalone step to be optimized**, but not the focus of our work. **Our primary focus was to handle both model and data heterogeneity in a challenging serverless decentralized FL setting** and **develop a theoretically solid framework to justify the proposed loss functions given being able obtain ideal synthetic data**. Our theoretical findings are versatile and can be applied to other distillation strategies, potentially including those suitable for high-dimensional data.
>
> Finally, we would like to **reiterate the contributions of our study**: we develop a theoretically solid framework to handle both model and data heterogeneity in a serverless decentralized FL scenario, and, empirically, we apply dataset distillation to generate synthetic data that fits our theory. The experimental result on benchmark datasets out-performs existing decentralized FL methods, which validates the effectiveness of DeSA.
>
> We believe our framework **lays a solid foundation for future exploration in data and model heterogeneous FL**, and we are hopeful that subsequent research will build on our theoretical insights to address a broader range of applications, including those involving high-dimensional real data.
> We appreciate the reviewer's insights and will highlight potential areas for future exploration.
>
> [1] Cazenavette G, Wang T, Torralba A, Efros AA, Zhu JY. Dataset distillation by matching training trajectories. InProceedings of the IEEE/CVF Conference on Computer Vision and Pattern Recognition 2022 (pp. 4750-4759).
>
> [2] Yu R, Liu S, Wang X. Dataset distillation: A comprehensive review. arXiv preprint arXiv:2301.07014. 2023 Jan 17.
>
> [3] Zhao B, Bilen H. Dataset condensation with distribution matching. InProceedings of the IEEE/CVF Winter Conference on Applications of Computer Vision 2023 (pp. 6514-6523).

---

### Official Review · Reviewer_xb5U · 2023-10-31

**Soundness:** 3 good
**Presentation:** 2 fair
**Contribution:** 3 good
**Rating:** 6
**Confidence:** 3

**Summary:**

The paper studied data and model heterogeneities in decentralized federated learning (FL), which is a serverless FL setting. In particular, the paper focused on the generalization, beyond personalization, of client models. The proposed method, DeSA, leverages synthetic anchors using data generation techniques to introduce two effective regularization terms for local training: anchor loss that matches the distribution of the client’s latent embedding with an anchor, and KD loss that enables clients learning from others. Experiments demonstrated the effectiveness of DeSA on intra- and inter-domain tasks.

**Strengths:**

1. The paper considered a complex setting where both data and model heterogeneities are present, which can be hard to tackle in general. New loss terms are introduced to deal with the heterogeneities, and data synthesis technique are used to avoid sharing real data. The approach is reasonable and justified.

2. The paper provided extensive experimental results to demonstrate the effectiveness of DeSA, which is compared against methods from both model heterogenous and homogeneous settings.

**Weaknesses:**

1. The motivation of considering generalization ability of client models on inter-domain tasks is not clear. In the model heterogenous setting, each client may process a model with a different architecture,  which is compatible with its own configuration. While the client can benefit from other clients’s data to train a personalized model, why does this model have to perform well on other clients’ tasks too? Other clients may not be able to acquire or deploy the model.

2. In experiments, from Table 2 it seems that data heterogeneity and model heterogeneity are correlated. That is, each dataset is assigned one model architecture. The results would be more interesting if both different datasets and models are assigned independently (by dividing a dataset into multiple clients).

**Questions:**

1. In DIGITS and OFFICE experiments, what is the number of clients? Is a client identified as a dataset?

2. How does DeSA perform in cases where each client has limited training data, e.g., a few samples per class? Can each model benefit more from the global synthetic dataset and KD?

3. Minor issues:
- In Equation (3), the definition of D_i^{Syn} involves summation over i.
- In Equation (6), L_{CE} is not used (but introduced right after the equation).
- First line of Section 5.3: FedSAB?

---

> ### Author Response · Authors · 2023-11-20
> **Justification of our motivation for federated mutual learning**
>
> We thank the reviewer for raising this concern. Let us think about an essential scenario: In 2020, hospitals from different regions want to collaboratively train a COVID-19 classification model. Due to quarantine, people cannot travel outside the local area, so the local hospital can only collect local cases with local strains of Covid viruses. If they train a model that only cares about intra-domain, this *personalized* model will suffer from a performance drop when the quarantine restriction is revoked and people from one region (with its local virus strains) will travel to other regions and use other hospitals’ models. Thus, we consider *intra* and *inter*-domain performances to be both important, and allowing each client model to generalize well on other client domains can potentially improve model robustness. Thus, In this work, we consider a different FL setting without sever and define it as  *decentralized federated mutual learning*, which shares the same motivation as the existing work [1].
>
> [1] Huang W, Ye M, Du B. Learn from others and be yourself in heterogeneous federated learning. InProceedings of the IEEE/CVF Conference on Computer Vision and Pattern Recognition 2022 (pp. 10143-10153).

---

> ### Author Response · Authors · 2023-11-20
> **Explanation of the model heterogeneous setup**
>
> We thank the reviewer for suggesting that we should split a dataset and assign different models to subsets. In fact, we implement this strategy in our CIFAR10C experiments, where we split Cifar10C into 57 clients (subsets) and randomly assigned different model architectures to them. The objective of Table 2 is to show that our method can obtain competitive inter and intra-domain performance under both data and model heterogeneous scenarios.

---

> ### Author Response · Authors · 2023-11-20
> **Clarification of the client setting for DIGITS and OFFICE experiments**
>
> For DIGITS and OFFICE experiments, we assume each dataset represents one client following [1].  We further clarify the setting in our revision.
>
> [1] Li X, JIANG M, Zhang X, Kamp M, Dou Q. FedBN: Federated Learning on Non-IID Features via Local Batch Normalization. InInternational Conference on Learning Representations 2020 Oct 2.

---

> ### Author Response · Authors · 2023-11-20
> **Explanation of how limited number of local data affects DeSA**
>
> We thank the reviewer for the great question. In Eq. 8, if we have limited local data, the bound will be dominated by $\alpha^{Syn}$ since $\alpha$ is small. Thus, the performance depends on the quality of global synthetic data. However, we would also like to point out limited local training data may affect the generation of synthetic data and further decrease its utility.

---

> ### Author Response · Authors · 2023-11-20
> **Correction of the typos**
>
> We thank the reviewer for the careful review and pointing out the typos. We have addressed the typos as follows. We have also done more careful proofreading in our revision.
> - There shouldn’t be a summation over i to N, and we have corrected it.
> - The L_{CE} description shouldn’t be there and we have removed it.
> - We corrected it into DeSA.

---

> ### Author Response · Authors · 2023-11-22
> **Thank you and look forward to your feedback to our rebuttal**
>
> Dear Reviewer xb5U,
>
> As the rebuttal deadline is approaching, we would like to know if our rebuttals have addressed your concerns and questions. Also, we have tried our best to reflect your brilliant feedback to our revision. We appreciate the opportunity to discussing during this stage and are delighted to address your further question if there is any. If you are satisfied with our response and revision, we are grateful if you could kindly re-consider the rating for DeSA.
>
> Best Regards,
>
> DeSA authors

---

### Official Review · Reviewer_VrA4 · 2023-11-01

**Soundness:** 2 fair
**Presentation:** 1 poor
**Contribution:** 2 fair
**Rating:** 3
**Confidence:** 4

**Summary:**

The paper explores decentralized federated learning, focusing on both data and model heterogeneity—a notably challenging context where traditional FedAVG and its derivatives fall short. The authors introduce a novel approach, DESA (Decentralized FL with Synthetic Anchors), which employs synthetic anchors to act as class-specific feature centers. To generate these synthetic anchors, the authors utilize randomly sampled feature extractors and optimize data points using the empirical maximum mean discrepancy (MMD) loss. Subsequently, each client is trained using anchor loss and knowledge distillation loss to combat data and model heterogeneity, respectively. Experimental validation is conducted on domain-shifted datasets: DIGITS, OFFICE, and CIFAR10c, where DESA shows superior performance.

**Strengths:**

1. The problem formulation is both rigorous and practically relevant.

2. Experimental evidence substantiates the efficacy of DESA on DIGITS, OFFICE, and CIFAR10c datasets.

**Weaknesses:**

1. Certain aspects of the paper remain ambiguous.
1-1. Equation (2) introduces an objective that encompasses all clients for defining inter-client loss. However, the decentralized nature of the problem implies that each client can communicate only with adjacent nodes, raising questions about the feasibility of this objective.
1-2 Equation (3) suffers from unclear terminology; specifically, the meaning of the representation (x∣y) is not explained. Additionally, the methodology for generating "randomly sampled feature extractors" is also unclear.
1-3. Equation (4) contains undefined notations, requiring clarification.
2. The paper touches upon privacy concerns arising from the sharing of synthetic data but fails to delve deep enough into this critical issue. Given the importance of privacy in federated learning algorithms, the authors should offer a more comprehensive discussion, preferably in the main text rather than relegating it to the appendix.
3. The theoretical results are primarily based on Ben-David et al. (2010), a fact highlighted in the appendix but missing from the main text. This could potentially weaken the paper’s contribution.
4. The paper struggles to bridge the gap between theoretical claims and empirical results. The presented theorems are contingent upon strong assumptions, and their relevance to the experimental findings is not intuitively obvious.

**Questions:**

ould you please clarify the issues listed under "Cons"?

---

> ### Author Response · Authors · 2023-11-20
> **Explanation of the ambiguous parts in the paper**
>
> > Clarification of client structure and the objective of Eq. 2
>
> We thank the reviewer for pointing out the typo of Eq. 2. The summation should be from 1 to N(C_i), and we have corrected it in our revision. As shown in Algorithm 1, our method is designed to work only to receive neighbor node information, and DeSA is designed for peer-to-peer decentralized network. Although we are working on a peer-to-peer decentralized network but the loss requires all nodes’ information, we can leverage the FastMix algorithm to aggregate all nodes’ information [1,2]. This method can aggregate all nodes' information via adjacent nodes’ communication at a linear speed. It is very common in fully decentralized optimization. In fact, our method can also work if each node can only receive neighbor nodes’ information, and we empirically show the feasibility in our CIFAR10C experiments by sampling neighboring clients.
>
> > Clarification of (x|y)
>
> We thank the reviewer for the clarification question. $\psi(x|y)$ means the comparison is conditional on label y , which allows us to distill synthetic data based on each class c.
>
> > Clarification of “randomly sampled feature extractors”
>
> We thank the reviewer for the clarification question. Using randomly sampled feature extractors allows us to perform empirical MMD loss as described in [3]. To implement it, we simply randomly initialize the model parameters for each round with PyTorch.
>
> > Clarification of Eq. 4
>
> We appreciate reviewer’s careful review. . We have removed the unused notation $P$ and add description of $K$ in our revision.
>
> [1] Ye H, Zhou Z, Luo L, Zhang T. Decentralized accelerated proximal gradient descent. Advances in Neural Information Processing Systems. 2020;33:18308-17.
>
> [2] Luo L, Ye H. Decentralized Stochastic Variance Reduced Extragradient Method. arXiv preprint arXiv:2202.00509. 2022 Feb 1.
>
> [3] Bo Zhao and Hakan Bilen. Dataset condensation with distribution matching. In Proceedings of the IEEE/CVF Winter Conference on Applications of Computer Vision, pp. 6514–6523, 2023.

---

> ### Author Response · Authors · 2023-11-20
> **Justification of privacy preservation in DeSA**
>
> We thank the reviewer for the suggestion. Since our main focus is to solve the challenging issue - decentralized federated learning with data and model heterogeneities, we decided to put more focus on how our method can utilize data distillation to solve the problem in our manuscript.
>
> We agree that showing the privacy guarantee is important, so we have discussed the empirical privacy guarantee in Section 3.2 and pointed the readers to our Appendix B and C for our privacy-preservation results. Specifically, we provide a DP version for synthetic data in our Appendix C. In addition, we have shown our method can defend against Membership Inference Attack (MIA) [1] in Appendix B. We share the concern that incorporating a comprehensive discussion into the main text might limit the space available for elaborating on the technical details essential for understanding our innovative method. Our priority is to ensure that the core aspects of our approach are communicated effectively and clearly. Other reviewers seem to like our current presentation. In the revised manuscript, we underscored the sentence in the main text that directs readers to the appendix.
>
>
>
> [1] Nicholas Carlini, Steve Chien, Milad Nasr, Shuang Song, Andreas Terzis, and Florian Tramer. Membership inference attacks from first principles. In 2022 IEEE Symposium on Security and Privacy (SP), pp. 1897–1914. IEEE, 2022a.

---

> ### Author Response · Authors · 2023-11-20
> **Discussion regarding Ben-David et al (2010)’s work**
>
> We would like to point out that Ben - David 2010 et al’s work was a general bound for learning on new domains. Our theory’s  contribution is not only applying Ben-David et al’s work to our scenario, but also connecting it to our main algorithm through the addition of terms like $f^{syn}$ and $f^{syn}_{KD}$.This addition introduces novel mathematical insights into the connection between labelling functions and the anchor and KD regularization losses.
>
> We also provide a novel stronger version of our theorem in Proposition 2, that holds under the mild conditions enforced by our algorithm, and other novel and useful Lemmas in the appendix that theoretically motivate our algorithm.
>
> Following your valuable suggestion, we have edited the reference for Ben-David et al. in our revision.

---

> ### Author Response · Authors · 2023-11-20
> **Discussion about the gap between theoretical claims and empirical results**
>
> We thank the reviewer for his interesting remark. However, we strongly believe that our theorem gives remarkable insights into the empirical success of our algorithm.
>
> - The only assumption that our theorem makes is the convergence in probability of the encoded target and source domains, which is empirically enforced by our anchor loss regularizer $L_{reg}$. This is a novel condition in theorem 1, that inspired our design of the empirical anchor regularization term in Equation (7).
>
> - It is of paramount importance to note that the previous work of Ben-David 2010 et al, whose shoulders we build Theorem 1 upon, has no connections to any practical algorithm at all. It was our novel contribution to develop a bound that incorporated empirical connections like the labelling function of synthetic data $f^{syn} $ and the Extended KD data $f^{syn}_{KD}$. These labelling functions connect deeply with our empirical anchor and KD regularization losses .
> Theorem 1 helps us predict the global performance of models using the component weights $\alpha$ given to the empirical regularization terms.
> We have further clarified these connections in the appendix of our revision.
>
> - Our newly added ablation studies clearly show that improving the quality and quantity of synthetic data ($D^{syn}$) boosts both inter and intra accuracy. Remarkably, Theorem 1 accurately predicted this outcome. In simpler terms, the theorem suggests that better synthetic data reduces the generalization error of labeling functions ($f^{syn}$ and $f^{syn}_{KD}$), tightening the upper bound on global generalization error. This alignment between theory and experiment underscores the reliability of Theorem 1 in predicting empirical outcomes.
>
> - The connection between the quality of synthetic data and the upper bound in Theorem 1 is further confirmed during experiments in Section C of the Appendix. The injection of noise reduces the quality of the global synthetic data $D_{syn}$, which thereby worsens the upper bound guarantee in Theorem 1. This conforms with the empirical observation of a drop in inter and intra-client accuracy.
>
> Therefore, we strongly believe that our theory is connected with the empirical side of our problem, and helps in explaining it’s results.

---

> ### Author Response · Authors · 2023-11-22
> **Thank you and look forward to your feedback to our rebuttal**
>
> Dear Reviewer VrA4,
>
> As the rebuttal deadline is approaching, we would like to know if our rebuttals have addressed your concerns and questions. Also, we have tried our best to reflect your brilliant feedback to our revision. We appreciate the opportunity to discussing during this stage and are delighted to address your further question if there is any. If you are satisfied with our response and revision, we are grateful if you could kindly re-consider the rating for DeSA.
>
> Best Regards,
>
> DeSA authors

---

> > ### Comment · Reviewer_VrA4 · 2023-11-23
> >
> > Thank you for your reply. I have reviewed all of your responses and will take them into account when determining my final score and in discussions with reviewers and AC.

---

### Official Review · Reviewer_2BZe · 2023-11-01

**Soundness:** 2 fair
**Presentation:** 3 good
**Contribution:** 3 good
**Rating:** 6
**Confidence:** 4

**Summary:**

Decentralized FL enables clients to own different local models and separately optimize local data. How can every client's local model learn generalizable representation is unknown. To address this question, This paper proposes a Decentralized FL technique by introducing Synthetic Anchors, as DESA. Authors leverage the synthetic anchors to implement 1) anchor loss that matches the distribution of the client's latent embedding with an anchor and 2) KD loss that enables clients learning from others. The proposed method doesn't presume access to real public or a global data generator.

**Strengths:**

1. The studied problem is novel and well motivated.
2. Distilling local synthetic anchor is interesting.
3. There are theoretical analysis of the proposed methods, in which the new generalization bound is better.
4. Figure 3 is interesting, jointly considering worst local accuracy and global accuracy. Experiment results show significant improvements of the proposed method.

**Weaknesses:**

1. The local synthetic anchor dataset iss shared. Thus, the privacy of the synthesized anchor should be considerred. Although the DP is used to protect synthetic anchor. But could this defend against recovering the raw data?
2. It would be better to conduct a more ablation study to decouple the effect of the sythetic anchor and the KD loss.

**Questions:**

See weaknesses.

---

> ### Author Response · Authors · 2023-11-20
> **Privacy guarantee for sharing synthetic data**
>
> We thank the reviewer for the clarification question. We agree the privacy of the synthetic data is important, and we provide a DP version for synthetic data in our Appendix C. In addition, we have shown our method can defend against Membership Inference Attack (MIA) [1] in Appendix B. Regarding raw data recovery, since we don’t share gradients among clients (we only share logits w.r.t. global synthetic data), the commonly used gradient inversion attacks [2,3] may not be able to recover original raw data.
>
> [1] Nicholas Carlini, Steve Chien, Milad Nasr, Shuang Song, Andreas Terzis, and Florian Tramer. Membership inference attacks from first principles. In 2022 IEEE Symposium on Security and Privacy (SP), pp. 1897–1914. IEEE, 2022a.
>
> [2] Geiping J, Bauermeister H, Dröge H, Moeller M. Inverting gradients-how easy is it to break privacy in federated learning?. Advances in Neural Information Processing Systems. 2020;33:16937-47.
>
> [3] Huang Y, Gupta S, Song Z, Li K, Arora S. Evaluating gradient inversion attacks and defenses in federated learning. Advances in Neural Information Processing Systems. 2021 Dec 6;34:7232-41.

---

> ### Author Response · Authors · 2023-11-20
> **More fine-grained ablation study on Reg loss and KD loss**
>
> We thank the reviewer for the constructive comment. Following your suggestion, we have performed a more fine-grained hyperparameter search on Reg loss and KD loss. Specifically, we search $\lambda_{KD}$ from {0, 0.1, 0.5, 1, 2} and $\lambda_{REG}$ from {0, 0.01, 0.02, 0.05, 0.1}, and the experimental results are updated in our revision. The findings are aligned with our original version that $\lambda_{KD}$ helps improve the inter-accuracy increases, and $\lambda_{REG}$ helps improve the intra-accuracy as well as the inter-accuracy within a certain range (observe the peak inter-accuracy at $\lambda_{REG}=0.01$).

---

> > ### Comment · Reviewer_2BZe · 2023-11-20
> > **Thanks for the responses**
> >
> > Based on the contributions of this work and responses, I'd like to keep my original scores.

---

### Official Review · Reviewer_R93g · 2023-11-06

**Soundness:** 3 good
**Presentation:** 3 good
**Contribution:** 3 good
**Rating:** 6
**Confidence:** 3

**Summary:**

This paper studies the problem of decentralized mutual learning. The challenges of decentralized mutual learning, other than the ordinary data non-iid issue, include model-heterogeneity and no server-coordination. This paper tackles this problem via constructing synthetic anchor data, whose information is shared across clients to bridge the large gap among data distributions. The paper further designs novel losses including regularization loss for representations of both anchor and true data; and a knowledge distillation loss to tackle model heterogeneity issue. Some theoretical insight is provided and numerical experiments on several benchmarks show convincing results.

**Strengths:**

Disclaimer: the reviewer is not very familiar with the anchor data generation in federated learning. Thus, I may not accurately assess the novelty of the technique proposed by this paper.

- the problem this paper considers is interesting and important. Features like no central server coordination and model heterogeneity make practical sense.

- the proposed algorithm is intuitive, has theoretical insight. And it seems to be also communication-efficient since only logits of anchor data require to be transmitted across clients.

- the overal presentation is very good, and I find enjoyable to read the paper.

- experimental results seem to be convincing.

**Weaknesses:**

- overall I find the designed model contains a lot of subtlety, as it is quite complex and contains many components. So it appears a bit difficult to probe what really works and what does not.

For example,

(a) how difficult is the data synthesis process (i.e. eq. 3) when the data is highly non-iid across clients. Since it basically minimize discrepancy between representations of local data and global data, does this process always successfully generate satisfactory anchor regardless of how data is partitioned? there is some visualization of synthetic anchor in appendix, but the quality of synthetic anchor still seems to be mysterious.

(b)  the losses are not dissected well enough so that readers can make sure each loss is orthogonal, and plays its desirable role. the losses are designed based on intuitive heuristics. However, what role does each loss exactly play is not clear enough. For example, the anchor loss defined in eq 4, is that a bit overlapping with what eq 3 (i.e. anchor data synthesis)? basically, if data is generated from eq 3, will eq 4 automatically be relatively small?

Basically, whether these losses are overlapping, and whether these losses have monotonic correlation, is difficult to determine.

- following up on the subtlety of the model components, the ablation studies for DESA is not comprehensive enough to help. the hyperparameters (e.g. $\lambda_{KD}$, $\lambda_{REG}$, and IPC) are not searched comprehensively. For example, the inter accuracy vs. $\lambda_{KD}$ is still monotonic with the three data points, and readers cannot grasp a full picture of the role of $\lambda_{KD}$ or KD loss.

**Questions:**

Please see weaknesses.

---

> ### Author Response · Authors · 2023-11-20
> **Explanation of the data synthesis and its quality**
>
> We thank the reviewer for the constructive question. Let us think about an extreme non-iid scenario where clients lack data in class c, our Eq. 2 intuitively suggests that client i, lacking samples in class c, cannot reasonably generate data samples with label c. In such instances, we adjust the interpolation ratio according to the class prior, thus deweighting client i on synthetic data class c and adding weight to the clients with abundant data in class i. Consequently, this implies client i will have a greater reliance on other clients’ data in class c, thereby addressing heterogeneity and enhancing overall generalizability.
>
> **Regarding the quality of synthetic data:**
> We hope to clarify that the purpose of data distillation is not to generate hight-fidelity images that look similar to the original image, but rather the ones keeping the similar level of utilities. Our generated distilled data share similar patterns as the ones presented in the original distribution-based dataset distillation paper [1] (see its Fig. 2).
>
> [1] Bo Zhao and Hakan Bilen. Dataset condensation with distribution matching. In Proceedings of the IEEE/CVF Winter Conference on Applications of Computer Vision, pp. 6514–6523, 2023.

---

> ### Author Response · Authors · 2023-11-20
> **Correction and explanation of the non-overlapped characteristic of Eq. 3 and Eq. 4**
>
> We apologize for the typo in Eq. 3 which may lead to misunderstanding. There should not be a summation of clients in Eq. 3, and we have corrected it in our revision.
>
> To answer your question regarding Eq. 3 and Eq. 4: No, Eq. 3 and Eq. 4 are not overlapped.
> First, we hope to correct the typo in Eq. 3 – there should not be a summation of clients in Eq.3 as we stated it as a local data distillation loss. Sorry for leading the misunderstanding. The objective of Eq. 3 is to distill *local synthetic data* by minimizing the MMD of synthetic data and real data; thus, it is used to update client i’s local synthetic data ($D^{syn}_i). Differently, Eq. 4 aims to update the feature extractor parameters using Supervised Contrastive Loss. Since the inputs of Eq. 4 is *global synthetic data*($D^{syn}$) (the interpolation of every client’s *local synthetic data*) instead of client i’s *local synthetic data* ($D^{syn}_i), minimizing Eq. 3 does not imply Eq. 4 will be relatively small.

---

> ### Author Response · Authors · 2023-11-20
> **The inter accuracy vs. $\lambda_{KD}$ is monotonic**
>
> Thanks for noticing the results, which is in fact align with our theoretical analysis. As shown in Theroem 1, if we have a large $\lambda_{KD}$, the error bound is dominated by the KD loss, which is for improving the inter accuracy purpose. However, a too large KD loss will hurt intra accuracy.

---

> ### Author Response · Authors · 2023-11-20
> **More fine-grained ablation study on Reg loss and KD loss**
>
> We thank the reviewer for the constructive comment. Following your suggestion, we have performed a more fine-grained hyperparameter search on Reg loss and KD loss. Specifically, we search $\lambda_{KD}$ from {0, 0.1, 0.5, 1, 2} and $\lambda_{REG}$ from {0, 0.01, 0.02, 0.05, 0.1}, and the experimental results are updated in our revision. The findings are aligned with our original version that $\lambda_{KD}$ helps improve the inter-accuracy increases, and $\lambda_{REG}$ helps improve the intra-accuracy as well as the inter-accuracy within a certain range (observe the peak inter-accuracy at $\lambda_{REG}=0.01$).

---

> ### Author Response · Authors · 2023-11-20
> **The effect of IPC**
>
> Finding the minimal amount synthetic data is not the focus of this work. We have replaced “minimal” with “small” in our revision for clarification. Following your suggestion, we have performed a more detailed hyperparameter search on IPC. Specifically, we search IPC from {5, 10, 20, 50, 100, 200} and the experimental results are updated in our revision.
>
> Overall, blindly increasing the IPC does not guarantee to obtain optimal intra- and inter-accuracy. It will cause the loss function to be dominated by the last 2 terms of Eq. 8, \textit{i.e.,} by synthetic data rather than minimizing the empirical risk of classification on cross-entropy loss. However, synthesizing larger number of synthetic data may degrade its quality, and the sampled batch for $\mathcal{L}_{REG}$ may fail to capture the distribution.

---

> ### Author Response · Authors · 2023-11-22
> **Thank you and look forward to your feedback for our rebuttal**
>
> Dear Reviewer R93g,
>
> As the rebuttal deadline is approaching, we would like to know if our rebuttals have addressed your concerns and questions. Also, we have tried our best to reflect your brilliant feedback to our revision. We appreciate the opportunity to discussing during this stage and are delighted to address your further question if there is any. If you are satisfied with our response and revision, we are grateful if you could kindly re-consider the rating for DeSA.
>
> Best Regards,
>
> DeSA authors

---

### Author Response · Authors · 2023-11-20
**Overall response**

We thank the reviewers for their valuable comments for DeSA. The reviews and suggestions are extremely helpful to make DeSA into a better shape. We briefly summarize our responses as follows:

**Strength**

We thank the reviewers for the positive feedbacks, which are:
- The problem setting with no central server and model and data heterogeneity is interesting and of practical value. (R93g, 2BZe, VrA4, xb5U, phME)
- The method is interesting, intuitive and has theoretical insight. (R93g, 2BZe, xb5U, phME)
- The method is communication efficient. (R93g)
- The overall presentation is good. (R93g, phME)
- Experiment results seem convincing. (R93g, 2BZe, VrA4, xb5U)
- Figure 3 is interesting. (2BZe)

**Justification**

> Justification of our motivation for federated mutual learning (xb5U, phME)

Reviewer xb5U and phME ask about the motivation for federated mutual learning. We share the same motivation as the existing work [1] and further provide a realistic example to explain the importance of handling both intra- and inter-performance.

[1] Huang W, Ye M, Du B. Learn from others and be yourself in heterogeneous federated learning. In Proceedings of the IEEE/CVF Conference on Computer Vision and Pattern Recognition 2022 (pp. 10143-10153).


> Discussion on Privacy Guarantee for DeSA (2BZe, VrA4)

- We highlight the discussion and differential privacy experiment on distilled local synthetic data we made in our original submission.

> Further interpretation of the theoretical analysis (VrA4, phME)
- We justify why the assumption is reasonable, interpret why the gap between theory and practice is small, and highlight how our empirical results are aligned with our theoretical results.

**Clarification**

> Explanation of the data synthesis and its quality (R93g)

Reviewer R93g asks about the data synthesis under highly non-iid scenario and the quality of synthetic data. We explain that we can adjust the interpolation ratio in the global synthetic data generation. Regarding the quality of synthetic data, we clarify that the purpose of data distillation is not to generate high-fidelity images that look similar to the original image, but rather the ones keeping the similar level of utilities.

> Explanation of how limited number of local data affects DeSA (xb5U)

We explain that with limited local data, the performance is determined by $\alpha^{Syn}$. However, the synthetic data quality caould also be affected.

> Correction and explanation of Eq. 3 (R93g, phME)

We correct the typo in Eq. 3, as local data synthesis only rely on local data.

Reviewer R93g asks about the overlap of Eq. 3 and Eq. 4, and we clarify that since they are two objective functions for updating local synthetic data and updating local feature extractor, they are not overlapped.

Reviewer phME asks for clarification of Eq. 3, and we respond that we have corrected the typo, and describe the class-balance guarantee and the privacy preservation. Using Eq. 3.

> Clarification on the communication cost of DeSA (phME)

We describe the communication overhead of DeSA and classical FL setting, and show that DeSA is communication efficient by sharing only logits w.r.t. Global synthetic data.

**Experiment**

> Fine-grained ablation study on Reg loss and KD loss (R93g, 2BZe)

We expand our ablation study by searching $\lambda_{KD}$ from {0, 0.1, 0.5, 1, 2} and $\lambda_{REG}$ from {0, 0.01, 0.02, 0.05, 0.1}. We have also updated the new results to the experiment section in our revision.

> The effect of IPC (phME)

We expand our ablation study by searching IPC from {5, 10, 20, 50, 100, 200}. We have also updated the new results to the experiment section in our revision.


We have carefully addressed the reviewers’ comments and have updated the manuscript accordingly. We kindly request the reviewers to evaluate our responses and revision, and we are happy to answer any further questions.

Best Regards,

DeSA authors

---

### Meta-Review · Area_Chair_2UAp · 2023-12-07

**Metareview:**

This paper studies decentralized federated learning with both data and model heterogeneity. The authors adopt the local data generalization techniques to boost local training. Then, two regularization techniques, anchor loss, and knowledge distillation loss, are adopted to solve the data and model heterogeneity, respectively. Extensive experiments demonstrate the effectiveness of the proposed approach.

Strengths:

(1)   The problem this paper considered is interesting and important. Decentralized federated learning is an emerging direction for real-world applications.

(2)   This paper is well written. The mathematical formulation of the proposed method is rigorous and clear.

(3)   The experiments are extensive to demonstrate the efficacy of the proposed approach.

Weaknesses:

(1) The gap between theoretical and empirical analysis of DeSA (e.g., the assumptions behind Theorem 1 are not well explained and verified, and the connection of generalization bounds and loss terms in DeSA is unclear). Furthermore, the technical challenges of the theoretical analysis are not clear.

(2) The experimental evaluation is unconvincing since the data sets in the experiments are not explicitly associated with the concerns regarding improving the generalization of a model on other clients.

(3) The trade-off analysis between inter- and intra-client generalization is not convincing (e.g., theoretically how it is related to the terms in Theorem 1, and empirically how the hyperparameters on the two regularization terms affect the trade-off?).

(4) The responses provided do not adequately address the privacy concern.

Even after the author's response and discussion with reviewers, the concerns remain unresolved. Therefore, I have to recommend rejection.

**Justification For Why Not Higher Score:**

N/A

**Justification For Why Not Lower Score:**

N/A

---

### Decision · Program_Chairs · 2024-01-16

Reject